# Provable convergence guarantees for black-box variational inference

**Justin Domke**
University of Massachusetts Amherst
domke@cs.umass.edu

**Guillaume Garrigos**
Université Paris Cité and Sorbonne Université, CNRS
Laboratoire de Probabilités, Statistique et Modélisation
F-75013 Paris, France
garrigos@lpsm.paris

**Robert Gower**
Center for Computational Mathematics
Flatiron Institute, New York
rgower@flatironinstitute.org

## Abstract

Black-box variational inference is widely used in situations where there is no proof that its stochastic optimization succeeds. We suggest this is due to a theoretical gap in existing stochastic optimization proofs—namely the challenge of gradient estimators with unusual noise bounds, and a composite non-smooth objective. For dense Gaussian variational families, we observe that existing gradient estimators based on reparameterization satisfy a quadratic noise bound and give novel convergence guarantees for proximal and projected stochastic gradient descent using this bound. This provides rigorous guarantees that methods similar to those used in practice converge on realistic inference problems.

## 1 Introduction

Variational inference tries to approximate a complex target distribution with a distribution from a simpler family [1, 2, 3, 4]. If $p(z, x)$ is a target distribution with latent variables $z \in \mathbb{R}^d$ and observed data $x \in \mathbb{R}^n$, and $q_w$ is a variational family with parameters $w \in \mathcal{W}$, the goal is to minimize the negative evidence lower bound (ELBO)

$$f(w) := \underbrace{- \mathop{\mathbb{E}}_{\mathsf{z} \sim q_w} \log p(\mathsf{z}, x)}_{l(w)} + \underbrace{\mathop{\mathbb{E}}_{\mathsf{z} \sim q_w} \log q_w(\mathsf{z})}_{h(w)}. \tag{1}$$

For subsequent discussion, we have decomposed the objective into the *free energy* $l$ and the *negative entropy* $h$. Minimizing $f$ is equivalent to minimizing the KL-divergence between $q_w$ to $p(\cdot|x)$, because $\mathrm{KL}(q_w \| p(\cdot|x)) = f(w) + \log p(x)$. Recent research has focused on "black box" variational inference, where the target distribution $p$ is sufficiently complex that one can only access it through evaluating probabilities (or gradients) at chosen points [5, 6, 7, 8, 9, 10, 11]. Crucially, one can still get stochastic *estimates* of the variational objective $f$ and of its gradient, and use these to optimize.

Still, variational inference can sometimes be unreliable [12, 13, 14, 15], and some basic questions remain unanswered. Most notably: does stochastic optimization of $f$ converge to a minimum of the objective? There has been various progress towards answering this question. One line of research seeks to determine if the variational objective $f$ has favorable structural properties, such as smoothness or (strong) convexity [13, 16, 17] (Sec. 2.1). Another line seeks to control the "noise" (variance or expected squared norm) of different gradient estimators [18, 19, 20, 21] (Sec. 2.2). However, few

full convergence guarantees are known. That is, there are few known cases where applying a given stochastic optimization algorithm to a given target distribution is known to converge at a given rate.

We identify two fundamental barriers preventing from analysing this VI problem as a standard stochastic optimization problem. First, the gradient noise depends on the parameters $w$ in a non-standard way (Sec. 2.3). This adds great technical complexity and renders many traditional stochastic optimization proofs inapplicable. Second, stochastic optimization theory typically requires that the (exact) objective function $f$ is Lipschitz continuous or Lipschitz smooth. But in our VI setting, under some fairly benign assumptions, the ELBO is neither Lipschitz continuous nor Lipschitz smooth.

We obtain non-asymptotic convergence guarantees for this problem, under simple assumptions.

> **Central contributions (Informal).** Suppose that the target model $\log p(\cdot, x)$ is concave and Lipschitz-smooth, and that $q_w$ is in a Gaussian variational family parameterized by the mean and a factor of the covariance matrix (Eq. 2). Consider minimizing the negative ELBO $f$ using either one of the two following algorithms:
>
> - a proximal stochastic gradient method, with the proximal step applied to $h$ and the gradient step applied to $l$, estimating $\nabla l(w)$ with a standard reparameterization gradient estimator (Eq. 5), and using a triangular covariance factor;
> - a projected stochastic gradient method, with the gradient applied to $f = l + h$, estimating $\nabla f(w)$ using either of two common gradient estimators (Eq. 7 or Eq. 9), with the projection done over symmetric and non-degenerate (Eq. 3) covariance factors
>
> Then, both algorithms converge with a $1/\sqrt{T}$ complexity rate (Cor. 12), or $1/T$ if we further assume that $\log p(\cdot, x)$ is *strongly* concave (Cor. 13).
>
> We also give a new bound on the noise of the "sticking the landing" gradient estimator, which leads to faster convergence when the target distribution $p$ is closer to Gaussian, up to *exponentially* fast convergence when it is exactly Gaussian (Cor. 14).

This is achieved through a series of steps, that we summarize below.

1. We analyze the *structural properties* of the problem. Existing results show that with a Gaussian variational family, if $-\log p(\cdot, x)$ is (strongly) convex or Lipschitz smooth, then so is the free energy $l$. This is for instance known to be the case for some generalized linear models, and we give a new proof of convexity and smoothness for some hierarchical models including hierarchical logistic regression (see Appendix 7.3). The remaining component of the ELBO, the neg-entropy $h$, is convex when restricted to an appropriate set. It is not smooth, but it was recently proved to be smooth over a certain non-degeneracy set.

2. We study the noise of three common *gradient estimators*. They do not satisfy usual noise bounds, but we show that they all satisfy a new quadratic bound (Definition 5). For the sticking-the-landing estimator, our bound formalizes the longstanding intuition that it should have lower noise when the variational approximation is strong (Thm. 4).

3. We identify and solve the key *optimization* challenges posed by the above issues via new convergence results for the proximal and projected stochastic gradient methods, when applied to objectives that are smooth (but not *uniformly* smooth) and with gradient estimators satisfying our quadratic bound.

## 1.1 Related work

Recently, Xu and Campbell [22] analyzed projected-SGD (stochastic gradient descent) for Gaussian VI using the gradient estimator we will later call $g_{ent}$ (7), with a particular rescaling. They show that, *asymptotically* in the number of observed data, their method converges locally with a rate of $1/\sqrt{T}$, under mild assumptions. Our results are less general in requiring convexity, but are non-asymptotic, apply with other gradient estimators, and give a faster $1/T$ rate with strong convexity.

Lambert et al. [23] introduce a VI-like SGD algorithm, derived from a discretisation of a Wasserstein gradient flow, and show it converges at a $1/T$ rate for the 2-Wasserstein distance when the log posterior is smooth and strongly concave. This line was continued by Diao et al. [24], who propose a

proximal-SGD method based on the decomposition $f = l + h$. They obtain a $1/\sqrt{T}$ rate when the log posterior is smooth and convex, and a $1/T$ rate when it is smooth and strongly convex. Unlike typical black-box VI algorithms used in practice, these algorithms require computing the Hessian of the posterior. We analyze more straighforward applications of SGD to the VI problem using standard gradient estimators. Under the same assumptions, our algorithms have the same rates for KL-divergence, which imply the same rates for 2-Wasserstein distance by of Pinsker's inequality and that the total-variation norm upper-bounds Wasserstein distance [25, Remark 8.2].

In concurrent work, Kim et al. [26], consider a proximal-SGD method similar to our approach in Sec. 6. They obtain a $1/T$ convergence rate, similar to what Cor. 12 gives when the log posterior is strongly concave. They also consider alternative parametrizations of the scale parameters that render the ELBO globally smooth, and obtain a nonconvex result: Under a relaxed version of the Polyak-Lojasiewicz inequality, they can guarantee a $1/T^4$ rate.

## 2 Properties of Variational Inference (VI) problems

Traditionally, the ELBO was optimized using message passing methods, which essentially assume that $p$ and $q$ are simple enough that block-coordinate updates are possible [2, 3, 4]. However, in the last decade a series of papers developed algorithms based on a "black-box" model where $p$ is assumed to be complex enough that one can only evaluate $\log p$ (or its gradient) at selected points $z$. The key observation is that even if $p$ is quite complex, it is still possible to compute *stochastic estimates* of the gradient of the ELBO, which can be deployed in a stochastic optimization algorithm [5, 6, 8, 9, 10].

This paper seeks rigorous convergence guarantees for this problem. We study the setting where the variational family is the set of (dense) multivariate Gaussian distributions, parameterized in terms of $w = (m, C)$, where $m$ is the mean and $C$ is a factor of the covariance, i.e.

$$q_w(z) = \mathcal{N}\left(z|m, CC^\top\right). \tag{2}$$

We optimize over $w \in \mathcal{W}$, where $\mathcal{W} = \{(m, C) : C \succ 0\}$ and will further require $C$ to be either symmetric or triangular. This does not really change the problem, since for a given covariance matrix $\Sigma$ there always exists a symmetric (or lower triangular) positive definite factor $C$ such that $CC^\top = \Sigma$.

Now, is it possible to solve this optimization problem? Without further assumptions, it is unlikely any guarantee is possible, because it is easy to encode NP-hard problems into this VI framework [27, 28]. We discuss below the assumptions we will make to be able to solve the problem.

### 2.1 Structural properties and assumptions

The properties of the free energy $l$ depend on the properties of the target distribution $p$. It is necessary to assume that $p$ is somehow "nice" to ensure the problem can be solved. Titsias and Lázaro-Gredilla [17, Proposition 1] showed that if $-\log p(\cdot, x)$ is convex, then $l$ is convex too. Challis and Barber [16, Sec 3.2] showed that if the likelihood $p(\cdot|z)$ is convex and the prior $p(z)$ is Gaussian, then $l$ is strongly-convex. Domke [13, Theorem 9] showed that if $-\log p(\cdot, x)$ is $\mu$-strongly convex, then $l$ is $\mu$-strongly convex as well, and that the constant is sharp. Similarly, Domke [13, Theorem 1] showed that if $\log p(\cdot, x)$ is $M$-smooth, then $l$ is also $M$-smooth, and that the constant is sharp.

In this paper we make two assumptions about the target distribution $p$: the negative log-probability $-\log p(\cdot, x)$ must be *convex* (or strongly convex), and *Lipschitz smooth*. Section 7.3 (Appendix) gives some example models where these assumptions are satisfied. For example, if the model is Bayesian linear regression, or logistic regression with a Gaussian prior, then $-\log p(\cdot, x)$ is smooth and strongly convex. In addition, if the target is a hierarchical logistic regression model, then $-\log p(\cdot, x)$ is smooth and convex.

Assumptions on $p$ also impact what a minimizer $w^* = (m^*, C^*)$ of $f(w)$ can look like. Intuitively, if the target $\log p(\cdot, x)$ is $\mu$-strongly concave, then we would expect the target distribution to be "peaky". This means that the optimal distribution would be close to a delta function centered at the MAP solution: $m^*$ would be close to some maximum of $\log p(\cdot, x)$ noted $\bar{m}$, and the covariance factor $C^*$ would not be too large. This intuition can be formalized: in this context we have $\|C^*\|^2 + \|m^* - \bar{m}\|_2^2 \leq d/\mu$ [13, Theorem 10]. Similarly, if $\log p(\cdot, x)$ is $M$-Lipschitz smooth, then we expect that the target is not too concentrated, so we might expect that the optimal covariance cannot not be too

small. Formally, it can be shown that the singular values of the covariance factor $C^*$ are greater than $1/\sqrt{M}$ [13, Theorem 7].

The properties of the neg-entropy $h$ are inherited from the choice of the variational family and do not depend on $p$. Since we consider a Gaussian variational family, $h$ is known in closed-form. To avoid some technical issues, we will restrict $h$ so that the covariance factor is positive definite, so $h(w)$ is equal to $-\ln \det C$ up to a constant if $C$ is positive definite (see Appendix 7.1), and to $+\infty$ otherwise. So $h$ inherits the properties of the negative log determinant, meaning it is a proper closed convex function whenever $C$ is symmetric or triangular (see Appendix 7.2). From an optimization perspective, it is natural to use a gradient-based algorithm. But unfortunately $h$ is not Lipschitz smooth on its domain, because its gradient can change arbitrarily quickly when the singular values of $C$ become small. However, it can be shown [13, Lemma 12] that $h$ is $M$-smooth over the following set of *non-singular* parameters (which contains the solution $w^*$, see the previous paragraph) given by

$$\mathcal{W}_M = \left\{ w = (m, C) : C \succ 0 \text{ and } \sigma_{\min}(C) \geq \frac{1}{\sqrt{M}} \right\}. \tag{3}$$

Instead of computing gradients, an optimizer might want to use the proximal operator of $h$, which can also be computed in closed form (see Sec. 4).

## 2.2 Gradient estimators

This paper considers gradient estimators based on the "path" method or "reparameterization". These assume some base distribution $s$ is known, together with some deterministic transformation $T_w$, such that the distribution of $T_w(u)$ is equal to $q_w$ when $u \sim s$. Then, we can write

$$\nabla_w \mathop{\mathbb{E}}_{z \sim q_w} \phi(z) = \mathop{\mathbb{E}}_{u \sim s} \nabla_w \phi(T_w(u)). \tag{4}$$

In the case of multivariate Gaussians, the most common choice is to use $s = \mathcal{N}(0, I)$ and $T_w(u) = Cu + m$, which we will consider in the rest of the paper.

We will consider three different gradient estimators based on the path-type strategy (Eq. 4). We will provide new noise bounds for each of them (all the proofs are in Appendix 8). Our analysis is mostly based on the following general result [20, Thm. 3].

**Theorem 1.** *Let $T_w(u) = Cu + m$ for $w = (m, C)$. Let $\phi : \mathbb{R}^d \to \mathbb{R}$ be $M$-smooth, suppose that $\phi$ is stationary at $\bar{m}$, and define $\bar{w} = (\bar{m}, 0)$. Then*

$$\mathop{\mathbb{E}}_{u \sim \mathcal{N}(0,I)} \|\nabla_w \phi(T_w(u))\|_2^2 \leq (d+1)M^2 \|m - \bar{m}\|_2^2 + (d+3)M^2 \|C\|_F^2 \leq (d+3)M^2 \|w - \bar{w}\|_2^2.$$

*Furthermore, the first inequality cannot be improved.*

Intuitively, this result says that the noise of a gradient estimator is lower when the scale parameter $C$ is small and the mean parameter $m$ is close to a stationary point.

The first estimator we consider is $\nabla l(w)$ only. It is given by taking $\phi = -\log p$ into Eq. 4, i.e.

$$\mathsf{g}_{\text{energy}}(u) := -\nabla_w \log p(T_w(u), x), \quad u \sim \mathcal{N}(0, I). \tag{5}$$

The next result gives a bound on the noise of this estimator, which is a direct consequence of Theorem 1. The second line uses Young's inequality to bound the noise in terms of (i) the distance of $w$ from $w^*$ and (ii) a constant determined by the distance of $w^*$ from some *fixed* parameters $\bar{w}$.

**Theorem 2.** *Suppose that $\log p(\cdot, x)$ is $M$-smooth and has a maximum (or stationary point) at $\bar{m}$, and define $\bar{w} = (\bar{m}, 0)$. Then, for every $w$ and every solution $w^*$ of the VI problem,*

$$\begin{aligned} \mathbb{E} \|\mathsf{g}_{\text{energy}}(u)\|_2^2 &\leq (d+3)M^2 \|w - \bar{w}\|_2^2 \\ &\leq 2(d+3)M^2 \|w - w^*\|_2^2 + 2(d+3)M^2 \|w^* - \bar{w}\|_2^2. \end{aligned} \tag{6}$$

Second, we consider an estimator of the gradient of the full objective $l + h$. It is obtained by simply taking the above estimator and adding the true (known) gradient of the neg-entropy, i.e.

$$\mathsf{g}_{\text{ent}}(u) := \mathsf{g}_{\text{energy}}(u) + \nabla h(w), \quad u \sim \mathcal{N}(0, I). \tag{7}$$

The noise of $g_{\text{ent}}$ can be bounded since it only differs from $g_{\text{energy}}$ by the deterministic quantity $\nabla h(w)$, and the fact that—provided $w \in \mathcal{W}_L$—the singular values of $w$ cannot be too small and so $\nabla h(w)$ cannot be too large. This is formalized in the following theorem, where again, the second line relaxes the result into a term based on the distance of $w$ from $w^*$ plus constants. (Another type of noise bound [29] for $g_{\text{ent}}$ was obtained by Kim et al. [21] for diagonal Gaussians and a variety of parameterizations of the covariance.)

**Theorem 3.** *Suppose that $\log p(\cdot, x)$ is $M$-smooth, that it is maximal at $\bar{m}$, and define $\bar{w} = (\bar{m}, 0)$. Then, for every $L > 0$, for every $w \in \mathcal{W}_L$ and every solution $w^*$ of the VI problem,*

$$
\begin{aligned}
\mathbb{E} \left\| g_{\text{ent}}(u) \right\|_2^2 &\leq 2(d+3)M^2 \left\| w - \bar{w} \right\|_2^2 + 2dL \\
&\leq 4(d+3)M^2 \| w - w^* \|^2 + 4(d+3)M^2 \| w^* - \bar{w} \|^2 + 2dL.
\end{aligned}
\tag{8}
$$

While $g_{\text{ent}}$ used the exact gradient of $h$, it may be beneficial to use a stochastic estimator $h$ instead. To derive our third estimator, write the gradient of the ELBO as[1]

$$
\nabla l(w) + \nabla h(w) = \nabla_w \mathbb{E}_{z \sim q_w} \left[ -\log p(z, x) + \log q_v(z) \right] \Big|_{v=w},
$$

where the parameters $v$ serve as a way to "hold $w$ constant under differentiation". This leads to our third estimator, called the "sticking the landing" (STL) gradient estimator

$$
g_{\text{STL}}(u) := g_{\text{energy}}(u) + [\nabla_w \log q_v(T_w(u))]_{v=w}, \quad u \sim \mathcal{N}(0, I).
\tag{9}
$$

Intuitively, we expect that $g_{\text{STL}}$ will tend to have lower noise than $g_{\text{ent}}$ when the posterior is well-approximated by the variational distribution. The reason is that, as observed by Roeder et al. [30], if $q_w(z)$ were a *perfect* approximation of $p(z|x)$, then the two terms in Eq. 9 would exactly cancel (for every $u$) and so the estimator would have zero variance. Below we formalize this intuition in what we believe is the first noise bound on $g_{\text{STL}}$.

**Theorem 4.** *Suppose that $\log p(\cdot, x)$ is $M$-smooth. Consider the residual $r(z) := \log p(z, x) - \log q_{w^*}(z)$ for any solution $w^*$ of the VI problem, assume that it has a stationary point $\hat{m}$, and define $\hat{w} = (\hat{m}, 0)$. Then $r$ is $K$-smooth for some $K \in [0, 2M]$, and for all $w \in \mathcal{W}_M$,*

$$
\begin{aligned}
\mathbb{E} \left\| g_{\text{STL}} \right\|_2^2 &\leq 8(d+3)M^2 \| w - w^* \|_2^2 + 2(d+3)K^2 \left\| w - \hat{w} \right\|_2^2 \\
&\leq 4(d+3)(K^2 + 2M^2) \| w - w^* \|_2^2 + 4(d+3)K^2 \| w^* - \hat{w} \|_2^2.
\end{aligned}
\tag{10}
$$

*Moreover, if $p(\cdot|x)$ is Gaussian then $K = 0$.*

When the target is Gaussian then $K = 0$ in Eq. 10, meaning that when $w = w^*$, the STL estimator has no variance. This is a distinguishing feature of $g_{\text{STL}}$, as opposed to $g_{\text{energy}}$ or $g_{\text{ent}}$.

## 2.3 Challenges for optimization

Three major issues bar applying existing analysis of stochastic optimization methods to our VI setting: 1) non-smooth composite objective 2) lack of uniform smoothness and 3) lack of uniformly bounded noise of the gradient estimator.

The first issue is due to the non-smoothness of neg-entropy function $h$. This means that under the benign assumption that the target $\log p$ is smooth, the full objective $l + h$ *cannot* be smooth, since a nonsmooth function plus a smooth function is always nonsmooth. This renders stochastic optimization proofs (e.g. those for the "ABC" conditions [29]) that do not use projections or proximal operators inapplicable.

One way to tackle this first issue would be to use a non-smooth proof technique, but these rely on having a uniform gradient noise bound (our third issue). Alternatively, one can overcome the non-smoothness of the neg-entropy function by either using a proximal operator, or projecting onto a set where $h$ is smooth, namely $\mathcal{W}_M$. This is the strategy we pursue.

---

[1]One could also build an estimator without holding $w$ constant in this way. However, in the case of Gaussian variational families, this turns out to be mathematically equivalent to using a closed form entropy because $\log q_w(T_w(u)) = -\frac{1}{2} \| u \|_2^2 - \frac{d}{2} \log(2\pi) - \log |C|$, which has the same gradient (independent of $u$) as $h(w)$.

Table 1: Table of known black-box VI properties relevant to optimization. In some cases there could be multiple valid $w^*$ or $\bar{w}$ in which case these results hold for all simultaneously.

| Description | Definition |
|---|---|
| Estimator for $\nabla l$ | $\mathrm{g_{energy}} = -\nabla_w \log p(T_w(u), x)$ |
| Estimator for $\nabla l + \nabla h$ | $\mathrm{g_{ent}} = -\nabla_w \log p(T_w(u), x) + \nabla h(w)$ |
| Estimator for $\nabla l + \nabla h$ | $\mathrm{g_{STL}} = -\nabla_w \log p(T_w(u), x) + \nabla_w \log q_v(T_w(u))|_{v=w}$ |
| Constraint set | $\mathcal{W}_M = \{(m, C) : C \succ 0, \ \sigma_{\min}(C) \geq 1/\sqrt{M}\}$ |
| Optimum | $w^* \in \operatorname{argmin}_w l(w) + h(w)$ |
| Stationary point of $l$ | $\bar{w} = (\bar{z}, 0)$ for any $\bar{z}$ that is a stationary point of $\log p(\cdot, x)$ |

| Condition on $-\log p(z, x)$ | Consequence |
|---|---|
| none | $h(w)$ is convex when $C$ is symmetric or triangular |
| | $h(w)$ is $M$-smooth over $\mathcal{W}_M$ |
| convex | $l(w)$ is convex |
| $\mu$-strongly convex | $l(w)$ is $\mu$-strongly convex |
| | $\|w^* - \bar{w}\|_2^2 \leq \frac{d}{\mu}$ |
| $M$-smooth | $l(w)$ is $M$-smooth |
| | $w^* \in \mathcal{W}_M$ |
| | $\mathrm{g_{energy}}, \mathrm{g_{ent}},$ and $\mathrm{g_{STL}}$ are quadratically bounded |

The second issue is that existing analyses for proximal/projected stochastic methods either rely on a uniform noise bound ([31, Cor. 3.6]) or uniform smoothness [32, 33, 34, 35, 36], both of which are not known to be true for VI. By uniform smoothness, we refer to the assumption that $\log p(T_w(u), x)$ is $M$–smooth for every $u$, with $M$ being independent of $u$. Instead, we can only guarantee that $\log p(T_w(u), x)$ is smooth in expectation, i.e. that $l(w)$ is smooth. Several works [9, Cond. 1][37, Thm. 1][38, Sec. 4][39, Assumption A1][40, Thm. 1][41, Assumption 3.2] assumed that the full objective $l + h$ is smooth, but we are not aware of cases where this holds in practice for VI and in our parameterization (Eq. 2) this *cannot* be true if $\log p$ is smooth.

The third issue is the lack of a uniform noise bound. Most non-smooth convergence guarantees within stochastic optimization [42] assume that the noise of the gradient estimator is uniformly bounded by a constant. But this does not appear to be true even under favorable assumptions—e.g. it is untrue if the target distribution is a standard Gaussian. The best that one can hope is that the gradient noise can be bounded by a quadratic that depends on the current parameters $w$, and—depending on the estimator—even this may only be true when the parameters are in the set $\mathcal{W}_M$. Thus, our main optimization contribution is to provide theoretical guarantees for stochastic algorithms under such a specific noise regime, which we make precise in the next definition.

**Definition 5.** Let $\phi$ be a differentiable function. We say that a random vector g is a **quadratically bounded estimator** for $\nabla \phi$ at $w$ with parameters $(a, b, w^*)$, if it is unbiased $\mathbb{E}[\mathrm{g}] = \nabla \phi(w)$ and if the expected squared norm is bounded by a quadratic function of the distance of parameters $w$ to $w^*$, i.e. $\mathbb{E}\|\mathrm{g}\|_2^2 \leq a\|w - w^*\|_2^2 + b$.

The noise bounds derived in Section 2.2 for the estimators $\mathrm{g_{energy}}, \mathrm{g_{ent}},$ and $\mathrm{g_{STL}}$ imply that they are *uniformly quadratically bounded estimators*. See Appendix 8.2 for a table with the corresponding constants $a$ and $b$.

## 3 Stochastic Optimization with quadratically bounded estimators

In this section we give new convergence guarantees for the Prox-SGD algorithm and the Proj-SGD algorithm with quadratically bounded gradient estimators. Because these may be of independent

interest, they are presented generically, without any reference to the VI setting. We specialize these results to VI in Section 4. For both algorithms, we present results assuming the problem to be strongly convex or just convex. All proofs for this section can be found in the Appendix (Sec. 9).

## 3.1 Stochastic Proximal Gradient Descent

Here we want to minimize a function which is the sum of two terms $l + h$, were both $l$ and $h$ are proper closed convex functions, and $l$ is smooth. For this we will use stochastic proximal gradient.

**Definition 6** (Prox-SGD). *Let* $w^0$ *be a fixed initial parameters and let* $\gamma_1, \gamma_2, \cdots$ *be a sequence of step sizes. The stochastic proximal gradient (Prox-SGD) method is given by*

$$w^{t+1} = \operatorname{prox}_{\gamma_t h} \left( w^t - \gamma_t g^t \right),$$

*where* $g^t$ *is a gradient estimator for* $\nabla l(w^t)$, *and the proximal operator is defined as*

$$\operatorname{prox}_{\gamma h}(w) := \operatorname*{argmin}_{v} h(v) + \frac{1}{2\gamma} \| w - v \|_2^2.$$

**Theorem 7.** *Let* $l$ *be a* $\mu$-*strongly convex and* $M$-*smooth function, and let* $\bar{w} = \operatorname{argmin}(l)$. *Let* $h$ *be a proper closed convex function, and let* $w^* = \operatorname{argmin}(l + h)$. *Let* $(w^t)_{t\in\mathbb{N}}$ *be generated by the Prox-SGD algorithm, with a constant stepsize* $\gamma \in \left(0, \min\{\frac{\mu}{2a}, \frac{1}{\mu}\}\right]$. *Suppose that* $g^t$ *is a quadratically bounded estimator (Def. 5) for* $\nabla l$ *with parameters* $(a, b, w^*)$. *Then,*

$$\mathbb{E} \left\| w^{T+1} - w^* \right\|_2^2 \le (1 - \gamma\mu)^T \left\| w^0 - w^* \right\|_2^2 + \frac{2\gamma}{\mu} \left( b + M^2 \left\| w^* - \bar{w} \right\|_2^2 \right). \tag{11}$$

*Alternatively, if we use the decaying stepsize* $\gamma_t = \min\left\{ \frac{\mu}{2a}, \frac{1}{\mu} \frac{2t+1}{(t+1)^2} \right\}$, *then*

$$\mathbb{E}\|w^T - w^*\|^2 \le \frac{16\lfloor a/\mu^2 \rfloor^2}{T^2}\|w^0 - w^*\|^2 + \frac{8}{\mu^2 T} \left( b + M^2 \|w^* - \bar{w}\|_2^2 \right). \tag{12}$$

*In both cases,* $T = \mathcal{O}(\epsilon^{-1})$ *iterations are sufficient to guarantee that* $\mathbb{E}\|w^T - w^*\|^2 \le \epsilon$.

The above theorem gives an anytime $1/T$ rate of convergence when the stepsizes are chosen based on the strong convexity and gradient noise constants $\mu$ and $a$. Note that we do not need to know precisely those constants: We show in Appendix 9.3 that using any constant step size proportional to $T/\log T$ leads to a $\log T/T$ rate.

**Theorem 8.** *Let* $l$ *be a proper convex and* $M$-*smooth function. Let* $h$ *be a proper closed convex function, and let* $w^* \in \operatorname{argmin}(l + h)$. *Let* $(w^t)_{t\in\mathbb{N}}$ *be generated by the Prox-SGD algorithm, with a constant stepsize* $\gamma \in (0, \frac{1}{M}]$. *Suppose that* $g^t$ *is a quadratically bounded estimator (Def. 5) for* $\nabla l$ *with parameters* $(a, b, w^*)$. *Then,*

$$\mathbb{E} \left[ f(\bar{w}^T) - \inf f \right] \le \gamma \left( a \frac{\| w^0 - w^* \|^2}{(1 - \theta^T)} + b \right),$$

*where* $\theta \overset{\text{def}}{=} \frac{1}{1 + 2a\gamma^2}$ *and* $\bar{w}^T \overset{\text{def}}{=} \frac{\sum_{t=1}^T \theta^{t+1} w^t}{\sum_{t=1}^T \theta^{t+1}}$. *In particular, if* $\gamma = \frac{1}{\sqrt{aT}}$, *then*

$$\mathbb{E} \left[ f(\bar{w}^T) \right] - \inf f \le \frac{1}{\sqrt{aT}} \left( 2a\|w^0 - w^*\|^2 + b \right) \quad \forall T \ge \max\left\{ \frac{M^2}{a}, 2 \right\}.$$

*Thus,* $T = \mathcal{O}(\epsilon^{-2})$ *iterations are sufficient to guarantee that* $\mathbb{E} \left[ f(\bar{w}^T) - \inf f \right] \le \epsilon$.

## 3.2 Stochastic Projected Gradient Descent

Here we want to minimize a function $f$ over a set of constraints $\mathcal{W}$, where $\mathcal{W}$ is a nonempty closed convex set and $f$ is a proper closed convex function which is differentiable on $\mathcal{W}$. To solve this problem, we will consider the stochastic projected gradient algorithm.

**Definition 9** (Proj-SGD). Let $w^0$ be some fixed initial parameter, and let $\gamma_1, \gamma_2, \cdots$ be a sequence of step sizes. The projected stochastic gradient (Proj-SGD) method is given by

$$w^{t+1} = \text{proj}_{\mathcal{W}} \left( w^t - \gamma_t g^t \right),$$

where $g^t$ is a gradient estimator for $\nabla f(w^t)$, and the projection operator is defined as

$$\text{proj}_{\mathcal{W}}(w) = \underset{v \in \mathcal{W}}{\text{argmin}} \, \|v - w\|_2^2.$$

Note that we do not require $f$ to be *smooth*, meaning that this setting is not a particular case of the one considered in Section 3.1. In fact, the arguments we use in the proofs for Proj-SGD are more closely related to stochastic *sub*gradient methods than stochastic gradient methods.

**Theorem 10.** *Let $\mathcal{W}$ be a nonempty closed convex set. Let $f$ be a $\mu$-strongly convex function, differentiable on $\mathcal{W}$. Let $w^* = \text{argmin}_{\mathcal{W}}(f)$. Let $(w^t)_{t \in \mathbb{N}}$ be generated by the Proj-SGD algorithm, with a constant stepsize $\gamma \in \left( 0, \min\{\frac{\mu}{2a}, \frac{2}{\mu}\} \right]$. Suppose that $g^t$ is a quadratically bounded estimator (Def. 5) for $\nabla f$ with parameters $(a, b, w^*)$. Then,*

$$\mathbb{E} \left\| w^T - w^* \right\|^2 \leq \left( 1 - \frac{\mu\gamma}{2} \right)^T \left\| w^0 - w^* \right\|^2 + \frac{2\gamma b}{\mu}. \tag{13}$$

*Alternatively, if we use the decaying stepsize $\gamma_t = \min \left\{ \frac{\mu}{2a}, \frac{2}{\mu} \frac{2t+1}{(t+1)^2} \right\}$, then*

$$\mathbb{E} \left[ \|w^T - w^*\|^2 \right] \leq \frac{32a}{\mu^2 T^2} \|w^0 - w^*\|^2 + \frac{16b}{\mu^2 T}. \tag{14}$$

*In both cases, $T = \mathcal{O}(\epsilon^{-1})$ iterations are sufficient to guarantee that $\mathbb{E}\|w^T - w^*\|^2 \leq \epsilon$.*

Note that Eq. 13 is a sum of two terms: one that decays exponentially in $T$ and one that decreases only when the stepsize $\gamma$ is sufficiently small. This has an important consequence: if one uses the gradient estimator $g_{\text{STL}}$ and the target distribution is exactly a Gaussian, then $b = 0$ (Appendix 8.2), meaning that the algorithm will converge at an *exponential* rate. This is similar to many results in the stochastic optimization literature showing faster rates hold when interpolation holds [35].

**Theorem 11.** *Let $\mathcal{W}$ be a nonempty closed convex set. Let $f$ be a convex function, differentiable on $\mathcal{W}$. Let $w^* \in \text{argmin}_{\mathcal{W}}(f)$. Let $(w^t)_{t \in \mathbb{N}}$ be generated by the Proj-SGD algorithm, with a constant stepsize $\gamma \in (0, +\infty)$. Suppose that $g^t$ is a quadratically bounded estimator (Def. 5) for $\nabla f$ at $w^t$ with constant parameters $(a, b, w^*)$. Then,*

$$\mathbb{E} \left[ f(\bar{w}^T) - \inf_{\mathcal{W}} f \right] \leq \frac{\gamma}{2} \left( a \frac{\left\| w^0 - w^* \right\|^2}{1 - \theta^T} + b \right).$$

*where $\theta \overset{\text{def}}{=} \frac{1}{1 + a\gamma^2}$ and $\bar{w}^T \overset{\text{def}}{=} \frac{\sum_{t=0}^{T-1} \theta^{t+1} w^t}{\sum_{t=0}^{T-1} \theta^{t+1}}$. Finally if $\gamma = \frac{\sqrt{2}}{\sqrt{aT}}$ and $T \geq 2$ then*

$$\mathbb{E} \left[ f(\bar{w}^T) \right] - \inf f \leq \frac{\sqrt{2a}}{\sqrt{T}} \|w^0 - w^*\|^2 + \frac{b}{\sqrt{2aT}}. \tag{15}$$

*Thus, $T = \mathcal{O}(\epsilon^{-2})$ iterations are sufficient to guarantee that $\mathbb{E} \left[ f(\bar{w}^T) - \inf f \right] \leq \epsilon$.*

## 4 Solving VI with provable guarantees

We now specialize the optimization results of the previous section to our VI setting. We aim to minimize the negative ELBO, $f : \mathbb{R}^d \times V \longrightarrow \mathbb{R} \cup \{+\infty\}$, which decomposes as $f = l + h$ where

$$l(w) = - \underset{\mathsf{z} \sim q_w}{\mathbb{E}} \log p(\mathsf{z}, x) \quad \text{and} \quad h(w) = \underset{\mathsf{z} \sim q_w}{\mathbb{E}} \log q_w(\mathsf{z}) \text{ if } C \succ 0, +\infty \text{ otherwise}, \tag{16}$$

and where $V$ is the vector space of matrices in which the covariance factors $C$ belong. Under the assumption that the log-target $\log p(\cdot, x)$ is $M$-smooth and concave, we propose two strategies to minimize this objective. All the proofs for this section can be found in Appendix 10.

1. Apply the Prox-SGD algorithm to the sum $l + h$, using a proximal step with respect to $h$, and a stochastic gradient step with respect to $l$. The gradient of $l$ will be estimated with $g_{energy}$ (5), which is globally quadratically bounded. The prox of $h$ admits a closed form formula if we choose that $V$ is the space of lower triangular matrices.

2. Apply the Proj-SGD algorithm to minimize $f$ over $\mathcal{W}_M$. The gradient of $f$ will be estimated using either $g_{ent}$ (7) or $g_{STL}$ (9), both of which are quadratically bounded on $\mathcal{W}_M$. The projection onto $\mathcal{W}_M$ admits a closed form formula if choose that $V$ is the space of symmetric matrices.

**Corollary 12** (Prox-SGD for VI). *Consider the VI problem where $q_w$ is a multivariate Gaussian distribution (Eq. 2) with parameters $w = (m, C) \in \mathbb{R}^d \times \mathcal{T}^d$, and assume that this problem admits a solution $w^*$. Suppose that $\log p(\cdot, x)$ is $M$-smooth and concave (resp. $\mu$-strongly concave). Generate a sequence $w^t$ by using the Prox-SGD algorithm (Def. 6) applied to $l$ and $h$ (Eq. 16), using $g_{energy}$ (5) as an estimator of $\nabla l$. Let the stepsizes $\gamma_t$ be constant and equal to $1/(\sqrt{a_{energy}T})$ (resp. be decaying as in Theorem 7 with $a_{energy} = 2(d+3)M^2$). Then, for a certain average $\bar{w}^T$ of the iterates, we have for $T \geq 2$ that*

$$\mathbb{E}\left[f(\bar{w}^T) - \inf f\right] = \mathcal{O}(1/\sqrt{T}) \quad (resp.\ \mathbb{E}\left[\|w^T - w^*\|_2^2\right] = \mathcal{O}(1/T)).$$

**Corollary 13** (Proj-SGD for VI). *Consider the VI problem where $q_w$ is a multivariate Gaussian distribution (Eq. 2) with parameters $w = (m, C) \in \mathbb{R}^d \times \mathcal{S}^d$, and assume that this problem admits a solution $w^*$. Suppose that $\log p(\cdot, x)$ is $M$-smooth and concave (resp. $\mu$-strongly concave). Generate a sequence $w^t$ by using the Proj-SGD algorithm (Def. 9) applied to the function $f = l + h$ (Eq. 16) and the constraint $\mathcal{W}_M$ (Eq. 3), using $g_{ent}$ (7) or $g_{STL}$ (9) as an estimator of $\nabla f$. Let the stepsizes $\gamma_t$ be constant and equal to $\sqrt{2/(aT)}$ (resp. be decaying as in Theorem 10) with $a = a_{ent} = 4(d+3)M^2$ or $a = a_{STL} = 24(d+3)M^2$. Then, for a certain average $\bar{w}^T$ of the iterates, we have for $T \geq 2$ that*

$$\mathbb{E}\left[f(\bar{w}^T) - \inf f\right] = \mathcal{O}(1/\sqrt{T}) \quad (resp.\ \mathbb{E}\left[\|w^T - w^*\|_2^2\right] = \mathcal{O}(1/T)).$$

**Corollary 14** (Proj-SGD for VI - Gaussian target). *Consider the setting of Corollary 13, in the scenario that $\log p(\cdot, x)$ is $\mu$–strongly concave, that we use the $g_{STL}$ estimator, and that we take a constant stepsize $\gamma_t \equiv \gamma \in \left(0, \min\{\frac{\mu}{2a_{STL}}, \frac{2}{\mu}\}\right]$. Assume further that $p(\cdot|x)$ is Gaussian. Then,*

$$\mathbb{E}\left[\|w^T - w^*\|_2^2\right] \leq \left(1 - \frac{\mu\gamma}{2}\right)^T \|w^0 - w^*\|_2^2.$$

Let us now discuss the practical implementation of these two algorithms (more details and an explicit implementation of the methods are given in Appendix 10.2). These require computing either the proximal operator of the neg-entropy $h$, or the projection onto the set of non-degenerate covariance factors $\mathcal{W}_M$. These can be computed [13] for every $w = (m, C)$ as:

- $\text{prox}_{\gamma h}(w) = (m, C + \Delta C)$, where $\Delta C$ is diagonal with $\Delta C_{ii} = \frac{1}{2}(\sqrt{C_{ii}^2 + 4\gamma} - C_{ii})$,

- $\text{proj}_{\mathcal{W}}(w) = (m, U\tilde{D}U^\top)$, where $C$ has the SVD $C = UDU^\top$ and $\tilde{D}_{ii} = \max\{D_{ii}, 1/\sqrt{M}\}$.

Our theory for the Prox-SGD algorithm formally assumes that the covariance factor $C$ lives in the vector space of lower-triangular matrices. This can be implemented by letting $C \in \mathbb{R}^{d \times d}$ and "clamping" the upper-triangular entries to zero throughout computation. Then the gradient $g_{energy}$ (5) can be computed using automatic differentiation, and the proximal operator can be computed as in the above equation. Or, one may choose $w = (m, c)$ where $c \in \mathbb{R}^{d \times (d+1)/2}$ is the lower-triangular entries of the matrix so $C = \text{tril}(c)$. Gradients with respect to $w = (m, c)$ can again be estimated using automatic differentiation (including the $\text{tril}$ operator), and the proximal operator can be computed by forming $C$, using the above formula, and then extracting the lower-triangular entries.

Similarly, our theory for the Proj-SGD algorithm formally assumes the covariance factor $C$ lives in the vector space of symmetric matrices. This can be implemented by letting $C \in \mathbb{R}^{d \times d}$, computing the gradient estimator $g_{ent}$ (7) or $g_{STL}$ (9) over the reals using automatic differentiation, and "symmetrizing" $C$ throughout computation. That is, set $C \leftarrow \frac{1}{2}C + \frac{1}{2}C^\top$ after applying each gradient update, right before projection. Or, one may choose $w = (m, c)$ where $c \in \mathbb{R}^{d \times (d+1)/2}$ is the lower-triangular entries of the matrix, so $C = \text{symm}(c)$, where $\text{symm}$ is a function similar to $\text{tril}$ except creating symmetric matrices. The gradient estimator $g_{ent}$ (7) or $g_{STL}$ (9) can be computed

using automatic differentiation (including the symm operator). In this case, to project one would first form $C$, then use the above formula, and then extract the lower-triangular matrices.

In terms of cost, computing $\log q_w(z)$ or $T_w(u)$ needs $\Theta(d^2)$ operations. Computing the prox of $h$ requires $d$ operations, but in contrast projecting onto $\mathcal{W}_M$ requires diagonalizing a symmetric matrix, which takes $\Theta(d^3)$ operations. We note that this is not necessarily an issue, because 1) eigenvalue decomposition has excellent computational constants in moderate dimensions; 2) computing the target $\log p(z, x)$ may itself require $\Theta(d^3)$ or more operations; 3) it is common to average a "minibatch" of gradient estimates, meaning each eigenvalue decomposition computation is amortized over many ELBO evaluations. Still, in high dimensions, when $\log p(z, x)$ is inexpensive, and small minibatches are used, computing such decomposition in each iteration could become a computational bottleneck. In this scenario, the Proj-SGD algorithm is clearly cheaper, with its $\Theta(d)$ cost for the proximal step.

## 5    Discussion

While this paper has focused on convergence with dense Gaussian variational distributions, the results in Sec. 3 apply more generally. It is conceivable that the necessary smoothness, convexity, and noise bounds could be established for other variational families, in which case these optimization results would provide "plug in" guarantees. We suspect this might be fairly easy to do with, e.g., elliptical distributions or location-scale families, using similar techniques as done with Gaussians. But it may be difficult to do so for broader variational families.

If $-\log p$ is strongly convex and smooth, another proof strategy is possible: Smoothness guarantees that $w^* \in \mathcal{W}_M$ and strong convexity guarantees that $w^* \in \{w : \|w - \bar{w}\|_2^2 \le d/\mu\}$. So one could do projected SGD, projecting onto the intersection of these two sets. The negative ELBO would be strongly convex and smooth over that set, and since $\|w - \bar{w}\|_2$ is bounded, gradient noise is upper-bounded by a constant, meaning classical projected SGD convergence rates that assume smoothness, strong convexity, and uniform gradient noise apply. However, projecting onto the intersection of those sets poses a difficulty and this uniform noise bound may be loose in practice.

Several variants of the gradient estimators we consider have been proposed to reduce noise, often based on control variates or sampling from a different base distribution [30, 37, 43, 44, 45, 46, 47, 48, 49, 50]. It would be interesting to establish gradient noise bounds for these estimators.

Various VI variants have been proposed that use *natural* gradient descent. It would also be interesting to seek convergence guarantees for these algorithms.

VI can be done with "score" or "reinforce" gradient estimators [6, 8], which use the identity

$$\nabla_w \mathbb{E}_{z \sim q_w} \phi(z) = \mathbb{E}_{z \sim q_w} \phi(z) \nabla_w \log q_w(z)$$

in place of how we used Eq. 4 for the "path" estimators we considered. While path estimators often seem to have lower variance, this is not always true: Take an unnormalized log-posterior $\phi(z)$ where $z$ is scalar. Now, define $\phi'(z) = \phi(z) + \sqrt{\epsilon} \sin(z/\epsilon)$, where $\epsilon$ is very small. Then, $\phi$ and $\phi'$ represent almost the same posterior, and score estimators for them will have almost the same variance. Yet, the derivative of the added term is $\cos(z/\epsilon)/\sqrt{\epsilon}$, so a path estimator for $\phi'$ will have much higher variance than one for $\phi$. This underlines why smoothness is essential for our guarantees—it excludes posteriors like $\phi'$. Future work further unravel when score estimators perform better than path estimators.

One attractive feature of VI is that data subsampling can often be used when computing gradients, decreasing the cost of each iteration. While we have not explicitly addressed this, our proofs can be easily generalized, at least for target distributions of the form $p(z, x) = p(z) \prod_{n=1} p(x_n|z)$. The only issue is to bound the noise of the subsampled estimator. While our gradient variance guarantees use Theorem 1 from [20], a more general result [20, Theorem 6] considers data subsampling. Very roughly speaking, with uniform data subsampling of 1 datum of a time, the gradient noise bounds in Thms. 2 and 3 would increase by a factor between 1 (no increase) and the number of data, depending on how correlated the data are—less correlation leads to a larger increase. (More precisely, the second term in Thm. 3 would not increase.) These increases would manifest as larger constants $a$ and $b$ in the quadratic bounds, after which exactly the same results hold. However, it is not obvious if the bound for $g_{\text{STL}}$ in Thm. 4 can be generalized in this way.

## Acknowledgements

We thank Kyurae Kim and an anonymous reviewer for pointing out an error in an earlier version of Lemma 32. This work was partially completed while the first two authors were visiting the Flatiron Institute and was supported in part by the National Science Foundation under Grant No. 2045900.

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

# Contents

# 6 Notations

We note $\mathcal{M}^d$ the space of $d \times d$ matrices, $\mathcal{S}^d$ the subspace of symmetric matrices, and $\mathcal{T}^d$ the subspace of lower triangular matrices. We use $\mathcal{W}_M := \{(m, C) \in \mathcal{W} : \sigma_{\min}(C) \geq \frac{1}{\sqrt{M}}\}$ to denote the set of parameters with positive definite covariance matrix whose singular values are greater or equal to $\frac{1}{\sqrt{M}}$.

| object | description |
|---|---|
| $x$ | observed variables |
| $z \in \mathbb{R}^d$ | latent variables |
| $p(z, x)$ | target distribution |
| $w = (m, C)$ | variational parameters ($m \in \mathbb{R}^d$ and $C \in \mathcal{M}^d, \mathcal{S}^d$ or $\mathcal{T}^d$) |
| $q_w(z) = \mathcal{N}(z|m, CC^\top)$ | Gaussian variational distribution |
| $l(w) = -\mathbb{E}_{z \sim q_w} \log p(z, x)$ | free energy |
| $h(w) = \mathbb{E}_{z \sim q_w} \log q(z)$ | negative entropy |
| $f(w) = l(w) + h(w)$ | negative ELBO |
| $w^* = \operatorname{argmin}_w l(w) + h(w)$ | optimal parameters |
| $\bar{w} = (\bar{z}, 0)$ | MAP parameters ($\bar{z} \in \operatorname{argmax}_z p(z, x)$) |
| $\mathbb{V}[g] = \operatorname{tr} \mathbb{C}[g]$. | variance of vector-valued random variables |

# 7 VI problems and their structural properties

This section contains the proofs of all the claims made in Section 2.

## 7.1 Modeling the VI problem

We recall and detail here the setup of our problem. Given a target distribution $p$ and observed data $x$, we want to solve the following Variational Inference problem

$$\min_{w \in \mathcal{W}} \operatorname{KL}(q_w \| p(\cdot|x)), \tag{VI}$$

where $q_w$ is a multivariate Gaussian variational family, whose density may be written as

$$q_w(z) = \frac{1}{\sqrt{(2\pi)^d \det(CC^\top)}} \exp\left(-\frac{1}{2}\|C^{-1}(z - m)\|^2\right), \quad w = (m, C). \tag{17}$$

We make the assumption (VI) has a non-degenerated solution, i.e. a solution for which the covariance matrix is invertible.

We will impose that our parameters have the form $w = (m, C) \in \mathbb{R}^d \times V$, where $V$ is some vector subspace of $\mathcal{M}^d$, the space of $d \times d$ matrices. On top of that, we will also want the covariance factor to be positive definite. In other words, we will impose that our parameters are taken from the set

$$\mathcal{W} := \{(m, C) \in \mathbb{R}^d \times V : C \succ 0\}.$$

For the sake of simplicity, the reader can assume for most of the paper that $V = \mathcal{M}^d$. But when coming to analyze our algorithms, we will see that we should take either $V = \mathcal{T}^d$ (the subspace of $d \times d$ lower triangular matrices) or $V = \mathcal{S}^d$ (the subspace of $d \times d$ symmetric matrices). This choice is dictated by two reasons: it guarantees that our problem remains convex (see Lemma 19), and it lowers the computational cost of our algorithms (see Section 10.2). Fortunately, requiring $C$ to be symmetric (or triangular) and positive definite does not change our problem, as stated in the next Lemma.

**Lemma 15** (Equivalent problems for triangular/symmetric covariance factors)**.** *Suppose that $\bar{w} \in \mathbb{R}^d \times \mathcal{M}^d$ is such that $q_{\bar{w}}$ minimizes $\operatorname{KL}(q_w \| p(\cdot|x))$ over $\mathbb{R}^d \times \mathcal{M}^d$. Then there exists $w^* = (m^*, C^*)$ such that $q_{\bar{w}}$ and $q_{w^*}$ have the same distribution, with $C^*$ being positive definite and symmetric (or lower triangular).*

*Proof.* Write $\bar{w} = (\bar{m}, \bar{C})$, and take $m^* = \bar{m}$. Let $\bar{\Sigma} = \bar{C}\bar{C}^\top$. Since we assumed that (VI) has a non-degenerated solution, we can assume without loss of generality that $\bar{\Sigma}$ is invertible. Therefore we can apply the Cholesky decomposition and write $\bar{\Sigma} = C^*(C^*)^\top$, where $C^*$ is lower triangular and positive definite. We can also define $C^* = (\bar{\Sigma})^{1/2}$ to obtain a symmetric and positive definite factor. $\qquad\square$

Once restricted to $\mathcal{W}$, our problem (VI) becomes equivalent to

$$\min_{w=(m,C)\in\mathbb{R}^d\times V} l(w) + h(w), \tag{18}$$

where

- $l(w) = -\mathbb{E}_{\mathsf{z}\sim q_w} \log p(\mathsf{z}, x)$ is the free energy ;
- $h(w) = \begin{cases} -\log\det C & \text{if } w \in \mathcal{W} \\ +\infty & \text{otherwise} \end{cases}$

This equivalence is mostly standard, but we state it formally for the sake of completeness.

**Lemma 16** (VI reformulated)**.** *Problems* (VI) *and* (18) *are equivalent.*

*Proof.* Let $w = (m, C) \in \mathcal{W}$, and let us show that $\mathrm{KL}(q_w \| p(\cdot|x)) = l(w) + h(w)$. By definition of the Kullback-Liebler divergence, and using $p(z|x) = p(z,x)/p(x)$, we can write

$$
\begin{aligned}
\mathrm{KL}(q_w \| p(\cdot|x)) &= \underset{\mathsf{z}\sim q_w}{\mathbb{E}} \log \frac{q_w(\mathsf{z})}{p(\mathsf{z}|x)} = \underset{\mathsf{z}\sim q_w}{\mathbb{E}} \log \frac{q_w(\mathsf{z})p(x)}{p(\mathsf{z}, x)} \\
&= \underset{\mathsf{z}\sim q_w}{\mathbb{E}} \log q_w(\mathsf{z}) - \underset{\mathsf{z}\sim q_w}{\mathbb{E}} \log p(\mathsf{z}, x) + \underset{\mathsf{z}\sim q_w}{\mathbb{E}} \log p(x) \\
&= \underset{\mathsf{z}\sim q_w}{\mathbb{E}} \log q_w(\mathsf{z}) + l(w) + \log p(x).
\end{aligned}
$$

If we note $H(X)$ the entropy of a random variable, we can see that $\mathbb{E}_{\mathsf{z}\sim q_w} \log q_w(\mathsf{z}) = -H(\mathsf{z})$ where $\mathsf{z} \sim q_w$. We can rewrite $\mathsf{z}$ via an affine transform as $\mathsf{z} = C\mathsf{u} + m$ with $\boldsymbol{u} \sim \mathcal{N}(0, I)$, so we can use [51, Section 8.6] to write (see also the arguments in [13])

$$H(\mathsf{z}) = H(\mathsf{u}) + \log|\det C|.$$

Here $H(\mathsf{u})$, the entropy of $\mathsf{u}$, is a constant equal to $(d/2)(1 + \log(2\pi)$. Moreover since $w \in \mathcal{W}$, we know that $C \succ 0$, which implies that $\log|\det C| = \log\det C$. Because $\log p(x)$ and $H(\mathsf{u})$ are both constants, we deduce that minimizing $\mathrm{KL}(q_w \| p(\cdot|x))$ over $w \in \mathcal{W}$ is equivalent to minimizing $l + h$, where $h(w) = -\log\det C$ if $w \in \mathcal{W}$ and $+\infty$ otherwise. $\qquad\square$

We emphasize that working with a triangular or symmetric representation for $C$ is not restrictive, and that doing computations in this setting is pretty straightforward, as we illustrate next.

**Lemma 17** (Computations with triangular/symmetric covariance factors)**.** *Let $V$ be a vector subspace of $\mathcal{M}^d$. Let $\phi : \mathbb{R}^d \times \mathcal{M}^d \to \mathbb{R}$ and let $\phi_V : \mathbb{R}^d \times V \to \mathbb{R}$ be the restriction of $\phi$ to $\mathbb{R}^d \times V$. Then:*

1. *If $\phi$ is convex then $\phi_V$ is convex.*

2. *If $\phi$ is differentiable then $\phi_V$ is differentiable, with $\nabla\phi_V(w) = \mathrm{proj}_{\mathbb{R}^d\times V}(\nabla\phi(w))$. In particular, $\|\nabla\phi_V(w)\| \leq \|\nabla\phi(w)\|$.*

3. *If $\phi$ is $M$-smooth over $\Omega \subset \mathbb{R}^d \times \mathcal{M}^d$, then $\phi_V$ is $M$-smooth over $\Omega \cap (\mathbb{R}^d \times V)$.*

4. *$\mathrm{proj}_{\mathcal{S}^d}(C) = (C + C^\top)/2$, and $\mathrm{proj}_{\mathcal{T}^d}(C) = \mathrm{tril}(C)$, where $\mathrm{tril}(C)$ sets to zero all the entries of $C$ above the diagonal.*

*Proof.*

1. This is immediate due to the fact that $V$ is convex, since it is a vector subspace of $\mathcal{M}^d$.

2. Let $\iota : \mathbb{R}^d \times V \to \mathbb{R}^d \times \mathcal{M}^d$, $\iota(m, C) = (m, C)$ be the canonical injection. Then we clearly have $\phi_V = \phi \circ \iota$. Moreover, $\iota$ is a linear application whose adjoint is $\iota^*$ is exactly the orthogonal projection $\mathrm{proj}_{\mathbb{R}^d \times V}$. So the computation of the gradient follows after applying the chain rule : $\nabla \phi_V(w) = \iota^*(\nabla \phi(\iota(w))) = \mathrm{proj}_{\mathbb{R}^d \times V}(\nabla \phi(w))$. As for the norm, it suffices to observe that the orthogonal projection is a linear operator with norm less or equal than $1$.

3. Using that the orthogonal projection is a contractive map gives

$$
\begin{aligned}
\|\nabla \phi_V(w) - \nabla \phi_V(w')\| &= \|\mathrm{proj}_{\mathbb{R}^d \times V}(\nabla \phi(w)) - \mathrm{proj}_{\mathbb{R}^d \times V}(\nabla \phi(w'))\| \\
&\leq \|\nabla \phi(w) - \nabla \phi(w')\| \leq M\|w - w'\|.
\end{aligned}
$$

4. This is a standard linear algebra result.

$\square$

## 7.2 Smoothness and convexity for VI problems

Now we state the main smoothness and convex properties for VI problems. We will see in our analysis that the smoothness properties often revolve around the following subset of $\mathcal{W}$

$$
\mathcal{W}_M := \{(m, C) \in \mathcal{W} : \sigma_{\min}(C) \geq \frac{1}{\sqrt{M}}\},
$$

where $\sigma_{\min}(C)$ refers to the smallest singular value of $C$.

**Lemma 18** (Smoothness for VI). *Assume that $\log p(\cdot, x)$ is $M$-smooth. Then*

1. *$l$ is $M$-smooth.*

2. *$h$ is differentiable over $\mathcal{W}$, with $\nabla h(w) = (0, -\mathrm{proj}_V(C^{-\top}))$ for all $w \in \mathcal{W}$.*

3. *$h$ is $M$-smooth over $\mathcal{W}_M = \{(m, C) \in \mathcal{W} : \sigma_{\min}(C) \geq \frac{1}{\sqrt{M}}\}$.*

*Proof.*

1. Combine [13, Theorem 1] with Lemma 17.3.

2. Combine the fact that the gradient of $-\log \det$ at a matrix $X$ is $-X^{-\top}$ with Lemma 17.2.

3. Combine [13, Lemma 12] with Lemma 17.3.

$\square$

**Lemma 19** (Convexity for VI). *Assume that $-\log p(\cdot, x)$ is convex (resp. $\mu$-strongly convex). Then*

1. *$l$ is convex (resp. $\mu$-strongly convex).*

2. *If $V = \mathcal{S}^d$ then $h$ is closed convex, $\mathcal{W}$ is convex, and $\mathcal{W}_M$ is convex and closed. Moreover,*

$$
\mathcal{W} = \left\{ (m, C) \in \mathbb{R}^d \times \mathcal{S}^d : \lambda_{\min}(C) > 0 \right\} \text{ and } \mathcal{W}_M = \left\{ (m, C) \in \mathbb{R}^d \times \mathcal{S}^d : \lambda_{\min}(C) \geq \frac{1}{\sqrt{M}} \right\}.
$$

3. *If $V = \mathcal{T}^d$ then $h$ is closed convex and $\mathcal{W}$ is convex. Moreover*

$$
\mathcal{W} = \{(m, C) \in \mathbb{R}^d \times \mathcal{T}^d : C \text{ has positive diagonal} \}.
$$

4. *If $V = \mathcal{M}^d$ then $h$ can be not convex.*

*Proof.*

1. Combine Titsias and Lázaro-Gredilla [17, Proposition 1] (resp. Domke [13, Theorem 9]) with Lemma 17.1.

2. Convexity of $h$ follows from [52, Example 7.13 and Theorem 7.17]. Convexity of $\mathcal{W}$ comes from the fact that it is essentially the set of symmetric positive definite matrices (which is equivalent to $\lambda_{\min}(C) > 0$). Let us now prove that $\mathcal{W}_M$ is convex and closed. First we notice that, because of symmetry, we can rewrite $\mathcal{W}_M$ as

$$\mathcal{W}_M = \{(m, C) \in \mathbb{R}^d \times \mathcal{S}^d \; : \; \lambda_{\min}(C) \geq \frac{1}{\sqrt{M}}\}.$$

This set is closed because $\lambda_{\min}$ is a continuous function. To see that $\mathcal{W}_M$ is convex we are going to use the fact that $\lambda_{\min}$ is a concave function. Take $C_1, C_2 \in \mathcal{W}_M, \alpha \in [0, 1]$ and write

$$
\begin{aligned}
\lambda_{\min}((1 - \alpha)C_1 + \alpha C_2) &= \min_{\|z\|=1} \langle((1 - \alpha)C_1 + \alpha C_2)z, z\rangle \\
&= \min_{\|z\|=1} (1 - \alpha)\langle C_1 z, z\rangle + \alpha\langle C_2 z, z\rangle \\
&\geq \min_{\|z\|=1} (1 - \alpha)\langle C_1 z, z\rangle + \min_{\|z\|=1} \alpha\langle C_2 z, z\rangle \\
&= (1 - \alpha)\lambda_{\min}(C_1) + \alpha\lambda_{\min}(C_2) \geq \frac{1}{\sqrt{M}}.
\end{aligned}
$$

3. Here $\mathcal{W} = \{(m, C) \in \mathbb{R}^d \times \mathcal{T}^d \; : \; C \succ 0\}$. Using Sylvester's criterion, it is easy to see that a lower triangular matrix is positive definite if and only if its diagonal has positive coefficients. In other words, $\mathcal{W} = \{(m, C) \in \mathbb{R}^d \times \mathcal{T}^d \; : \; C_{ii} > 0\}$, which is clearly convex. It remains to verify that $h$ is closed convex even if $\mathcal{W}$ is not closed. For all $w = (m, C) \in \mathbb{R}^d \times \mathcal{T}^d$, we can use the triangular structure of $C$ to write

$$h(w) = -\log \det C + \delta_{\mathcal{W}}(w) = \sum_{i=1}^{d} -\log C_{ii} + \delta_{(0,+\infty)}(C_{ii}),$$

where we used the notation $\delta_A$ for the indicator function of a set $A$, which is equal to zero on $A$ and $+\infty$ outside. Since $-\log(t) + \delta_{(0,+\infty)}(t)$ is a closed convex function, we see that $h$ is a separable sum of closed convex functions, and thus is itself closed and convex.

4. Let $d = 2$, let $I$ be the identity matrix, let $R = \left(\begin{smallmatrix} 0 & 1 \\ -1 & 0 \end{smallmatrix}\right)$ be a rotation matrix, and define $\psi : \mathbb{R} \to \mathbb{R}$ with $\psi(t) = h(I + tR)$. Observe that $\psi$ is well defined because $I + tR \succ 0$ for all $t \in \mathbb{R}$. On the one hand, if $h$ was convex, then $\psi$ would be convex too, as it is a composition of a convex function with the affine function $t \mapsto I + tR$. On the other hand, we can compute explicitly that $\psi(t) = -\log(1 + t^2)$, which is not convex.

$\square$

**Lemma 20** (Solutions of VI). *Assume that $\log p(\cdot, x)$ is $M$-smooth, and that $V = \mathcal{M}^d, \mathcal{S}^d$ or $\mathcal{T}^d$. Then $\mathrm{argmin}(f) \subset \mathcal{W}_M := \{(m, C) \in \mathcal{W} : \sigma_{\min}(C) \geq \frac{1}{\sqrt{M}}\}$.*

*Proof.* If $w^* = (m^*, C^*) \in \mathrm{argmin}(f)$, then $w^*$ is in the domain of $h$, which is $\mathcal{W}$ by definition of $h$. So it remains to prove that $\sigma_{\min}(C^*) \geq 1/\sqrt{M}$. When $V = \mathcal{M}^d$, this is proved in [13, Theorem 7]. Suppose now that $V = \mathcal{S}^d$, meaning that $w^*$ is a minimizer of $f$ over $\mathcal{W} \subset \mathbb{R}^d \times \mathcal{S}^d$. We see that $w^*$ must also be a minimizer of $f$ over $\mathbb{R}^d \times \mathcal{M}^d$, because if there was a better solution $\hat{w} = (\hat{m}, \hat{C})$ with $\hat{C} \in \mathcal{M}^d$, we could consider $(\hat{C}\hat{C}^\top)^{1/2} \in \mathcal{S}^d$ which would itself be a better solution than $C^*$, which is a contradiction. So we can apply [13, Theorem 7] to $w^*$ to conclude that $\sigma_{\min}(C^*) \geq 1/\sqrt{M}$. If $V = \mathcal{T}^d$, we can use the same argument, using this time the Cholesky decomposition of $\hat{C}\hat{C}^\top$. $\square$

**Lemma 21.** *Let $w = (m, C)$ with $C$ invertible. Then $\log q_w$ is $\sigma_{\min}(C)^{-2}$-smooth.*

*Proof.* According to the definition of $q_w$ (see (17)), the Hessian of $\log q_w$ is $\nabla_z^2(\log q_w)(z) = -(CC^\top)^{-1}$, and so $\|\nabla_z^2(\log q_w)(z)\| = \sigma_{\max}((CC^\top)^{-1}) = \sigma_{\min}(C)^{-2}$. $\square$

## 7.3 A case study : linear models

Table 2: Table of models

| Model | Smoothness constant | Strong convexity constant | Convex |
|---|---|---|---|
| Bayesian linear regression ($\Sigma = I$) | $1 + \frac{1}{\sigma^2}\sigma_{\max}(A)^2$ | $1 + \frac{1}{\sigma^2}\sigma_{\min}(A)^2$ | yes |
| Logistic regression ($\Sigma = I$) | $1 + \frac{1}{4}\sigma_{\max}(A)^2$ | 1 | yes |
| Heirarchical logistic regression | exists | n/a | yes |

**Definition 22.** A twice differentiable function $f$ is $M$-smooth if $-MI \preceq \nabla^2 f \preceq MI$.

**Definition 23.** A twice differentiable function $f$ is $\mu$-strongly convex $\nabla^2 f \succeq \mu I$.

**Theorem 24.** *Take a generic i.i.d. linear model defined as* $p(z, x) = p(z) \prod_{n=1}^{N} p(x_n | z, a_n)$, *where* $p(z) = \mathcal{N}(z | 0, \Sigma)$ *and*

$$p(x_n | z, a_n) = \exp\left(-\phi\left(z^\top a_n, x_n\right)\right)$$

*for some function $\phi$. Suppose that the second derivative of $\phi$ with respect to its first (scalar) argument is bounded by $\theta_{\min} \leq \phi'' \leq \theta_{\max}$. Let*

$$\mu \overset{\text{def}}{=} \lambda_{\min}\left(\Sigma^{-1} + \theta_{\min}AA^\top\right)$$

$$\beta \overset{\text{def}}{=} \lambda_{\max}\left(\Sigma^{-1} + \theta_{\min}AA^\top\right),$$

*where $A$ is a matrix with $a_n$ in the $n$-th column. Then*

1. *$-\log p(z, x)$ is $M$-smooth over $z$ for $M = \max\left(|\mu|, |\beta|\right)$.*

2. *If $\mu \geq 0$ then $-\log p(z, x)$ is convex over $z$*

3. *If $\mu > 0$ then $-\log p(z, x)$ is $\mu$-strongly convex over $z$.*

*Proof.* It is easy to show that

$$\nabla_z^2 \phi(z^\top a_n, x_n) = -\phi''(z^\top a_n, x_n)a_n a_n^\top$$

from which it follows that

$$-\nabla_z^2 \log p(z, x) = \Sigma^{-1} + \sum_n \phi''(z^\top a_n, x_n)a_n a_n^\top.$$

Now, take some arbitrary vector $v$. We have that

$$
\begin{aligned}
v^\top\left(-\nabla_z^2 \log p(z, x)\right)v &= v^\top \Sigma^{-1} v + \sum_n b_n \left\|v^\top a_n\right\|^2 \\
&\geq v^\top \Sigma^{-1} v + \theta_{\min}\sum_n \left\|v^\top a_n\right\|^2 \\
&= v^\top\left(\Sigma^{-1} + \theta_{\min}\sum_n a_n a_n^\top\right)v, \\
&= v^\top\left(\Sigma^{-1} + \theta_{\min}AA^\top\right)v,
\end{aligned}
$$

Similarly, we have that

$$
\begin{aligned}
v^\top\left(-\nabla_z^2 \log p(z, x)\right)v &= v^\top \Sigma^{-1} v + \sum_n b_n \left\|v^\top a_n\right\|^2 \\
&\leq v^\top \Sigma^{-1} v + \theta_{\max}\sum_n \left\|v^\top a_n\right\|^2 \\
&= v^\top\left(\Sigma^{-1} + \theta_{\max}\sum_n a_n a_n^\top\right)v, \\
&= v^\top\left(\Sigma^{-1} + \theta_{\max}AA^\top\right)v.
\end{aligned}
$$

Putting both the above results together, we have that

$$\Sigma^{-1} + \theta_{\min} AA^\top \preceq -\nabla_z^2 \log p(z, x) \preceq \Sigma^{-1} + \theta_{\max} AA^\top.$$

This implies that all eigenvalues of $-\nabla_z^2 \log p$ are in the interval $[\mu, \beta]$. All the claimed properties follow from this:

- For any eigenvalue $\lambda$ of $\nabla_z^2 \log p(z, x)$, we know that $\left|\lambda_i(\nabla_z^2 \log p(z, x))\right| \leq M = \max(|\mu|, |\beta|)$. Thus, $\log p(z, x)$ is $M$-smooth over $z$.

- If $\mu \geq 0$, this means all eigenvalues of $-\nabla_z^2 \log p(z, x)$ are positive, which implies that $-\log p(z, x)$ is convex.

- If $\mu > 0$, then all eigenvalues of $-\nabla_z^2 \log(z, x)$ are bounded below by $\alpha$, which impiles that $-\log p(z, x)$ is $\mu$-strongly convex.

$\qquad\square$

**Corollary 25.** *Take a linear regression model with inputs $a_n \in \mathbb{R}^d$ and outputs $x_n \in \mathbb{R}$. Let $p(z) = \mathcal{N}(z|0, \Sigma)$ and $p(x_n|z) = \mathcal{N}\left(x|z^\top a_n, \sigma^2\right)$ for some fixed $\sigma$. Then $-\log p(z, x)$ is $\mu$-strongly convex for $\mu = \lambda_{\min}\left(\Sigma^{-1} + \frac{1}{4}AA^\top\right)$ and $M$-smooth for $M = \lambda_{\max}(\Sigma^{-1} + \frac{1}{4}AA^\top)$.*

*Proof.* In this case we have that

$$\begin{aligned}
\phi(z^\top a_n, x_n) &= -\log p(x_n|z) \\
&= \frac{1}{2\sigma^2}\left(x_n - z^\top a_n\right)^2 - \frac{1}{2}\log(2\pi\sigma^2).
\end{aligned}$$

Or, more abstractly,

$$\begin{aligned}
\phi(\alpha, \beta) &= \frac{1}{2\sigma^2}(\alpha - \beta)^2 - \frac{1}{2}\log(2\pi\sigma^2). \\
\phi'(\alpha, \beta) &= \frac{1}{\sigma^2}(\alpha - \beta) \\
\phi''(\alpha, \beta) &= \frac{1}{\sigma^2}
\end{aligned}$$

Thus we have that $\theta_{\min} = \theta_{\max} = \frac{1}{\sigma^2}$. It follows from Theorem 24 that $-\log p(z, x)$ is $\mu$-strongly convex for $\mu = \lambda_{\min}\left(\Sigma^{-1} + \frac{1}{\sigma^2}AA^\top\right)$ and $M$-smooth for $M = \lambda_{\max}\left(\Sigma^{-1} + \frac{1}{\sigma^2}AA^\top\right)$.

In the particular case where $\Sigma = I$, we get that $\mu = \lambda_{\min}\left(I + \frac{1}{\sigma^2}AA^\top\right) = 1 + \frac{1}{\sigma^2}\sigma_{\min}(A)^2$ and $M = \lambda_{\max}(1 + \frac{1}{\sigma^2}AA^\top) = 1 + \frac{1}{\sigma^2}\sigma_{\max}(A)^2$. $\qquad\square$

**Corollary 26.** *Take a logistic regression model with inputs $a_n \in \mathbb{R}^d$ and outputs $x_n \in \{-1, +1\}$. Let $p(z) = \mathcal{N}(z|0, \Sigma)$ and $p(x_n = 1|z) = \mathrm{Sigmoid}(x_n z^\top a_n) = 1/(1 + \exp(-x_n z^\top a_n))$. Then $-\log p(z, x)$ is $\mu$-strongly convex for $\mu = \lambda_{\min}\left(\Sigma^{-1}\right)$ and $M$-smooth for $M = \lambda_{\max}(\Sigma^{-1} + \frac{1}{4}AA^\top)$.*

*Proof.* In this case we have that

$$\begin{aligned}
\phi(z^\top a_n, x_n) &= -\log p(x_n|z) \\
&= \log\left(1 + \exp(-x_n z^\top a_n)\right).
\end{aligned}$$

Or, more abstractly,

$$\begin{aligned}
\phi(\alpha, \beta) &= \log\left(1 + \exp(-\alpha\beta)\right) \\
\phi'(\alpha, \beta) &= -\frac{\beta}{1 + \exp(\alpha\beta)} \\
\phi''(\alpha, \beta) &= \frac{\beta^2 \exp(\alpha\beta)}{(1 + \exp(\alpha\beta))^2}
\end{aligned}$$

The second derivative is non-negative and goes to zero if $\alpha \to -\infty$ or $\alpha \to +\infty$. It is maximized at $\alpha = 0$ in which case $\phi'' = \frac{1}{4}\beta^2$. But of course, in this model, the second input is $\beta = x_n \in \{-1, +1\}$, meaning $\beta^2 =$ always. Thus we have that $\theta_{\min} = 0$ and $\theta_{\max} = 1$. It follows from Theorem 24 that $-\log p(z, x)$ is $\mu$-strongly convex for $\mu = \lambda_{\min}\left(\Sigma^{-1}\right)$ and $M$-smooth for $M = \lambda_{\max}\left(\Sigma^{-1} + \frac{1}{4}AA^\top\right)$.

In the particular case where $\Sigma = I$, we get that $\mu = 1$ and $M = \lambda_{\max}(1 + \frac{1}{4}AA^\top) = 1 + \frac{1}{4}\sigma_{\max}(A)^2$. $\qquad\square$

**Theorem 27.** *Take a heirarchical model of the form* $p(\theta, z, x) = p(\theta) \prod_i p(z_i|\theta)p(x_i|\theta, z_i)$, *where all log-densities are twice-differentiable. Then,* $\log p(\theta, z, x)$ *is smooth joinly over* $(\theta, z)$ *if and only if the spectral norm of the second derivative matrices* $\nabla_{\theta\theta^\top} \log p(\theta, z, x)$, $\nabla_{\theta z_i^\top} \log p(\theta, z_i, x_i)$, *and* $\nabla_{z_i z_i^\top} \log p(\theta, z_i, x_i)$ *are all bounded uniformly over* $(\theta, z)$.

*Proof.* The following conditions are all equivalent:

1. $\log p(\theta, z, x)$ is smooth as a function of $(\theta, z)$

2. $\left\|\nabla^2 \log p(\theta, z, x)\right\|_2$ is bounded above (uniformly over $(\theta, z)$, where $\nabla^2$ denotes the Hessian with respect to $(\theta, z)$ and $\|\cdot\|_2$ denotes the spectral norm.)

3. $\left\|\nabla^2 \log p(\theta, z, x)\right\|$ is bounded above (uniformly over $(\theta, z)$, where $\nabla^2$ denotes the Hessian with respect to $(\theta, z)$ and $\|\cdot\|$ denotes a block infinity norm on top of spectral norm inside each block.

4. $\left\|\nabla^2_{\theta\theta^\top} \log p(\theta, z, x)\right\|_2$, $\left\|\nabla^2_{\theta z_i^\top} \log p(\theta, z, x)\right\|_2$, and $\left\|\nabla^2_{z_i z_j^\top} \log p(\theta, z, x)\right\|_2$ are all bounded above (uniformly over $(\theta, z)$)

5. $\left\|\nabla^2_{\theta\theta^\top} \log p(\theta, z, x)\right\|_2$, $\left\|\nabla^2_{\theta z_i^\top} \log p(\theta, z_i, x_i)\right\|_2$, and $\left\|\nabla^2_{z_i z_j^\top} \log p(\theta, z, x)\right\|_2$ are all bounded above (uniformly over $(\theta, z)$)

6. $\left\|\nabla^2_{\theta\theta^\top} \log p(\theta, z, x)\right\|_2$, $\left\|\nabla^2_{\theta z_i^\top} \log p(\theta, z_i, x_i)\right\|_2$, and $\left\|\nabla^2_{z_i z_i^\top} \log p(\theta, z, x)\right\|_2$ are all bounded above (uniformly over $(\theta, z)$)

7. $\left\|\nabla^2_{\theta\theta^\top} \log p(\theta, z, x)\right\|_2$, $\left\|\nabla^2_{\theta z_i^\top} \log p(\theta, z_i, x_i)\right\|_2$, and $\left\|\nabla^2_{z_i z_i^\top} \log p(\theta, z_i, x_i)\right\|_2$ are all bounded above (uniformly over $(\theta, z)$)

$\qquad\square$

**Corollary 28.** *Take a hierarchical logistic regression model defined by* $p(\theta, z, x) = p(\theta) \prod_i p(z_i|\theta) \prod_j p(x_{ij}|z_i)$ *where* $\{\theta, z_i\} \in \mathbb{R}^d$, $x_{ij} \in \{-1, +1\}$, *and*

$$
\begin{aligned}
\theta &\sim \mathcal{N}(0, \Sigma) \\
z_i &\sim \mathcal{N}(\theta, \Delta) \\
x_{ij} &\sim p(x_{ij}|z_i) \\
p(x_{ij} = 1|z_i) &= \mathrm{Sigmoid}(x_{ij} z_i^\top a_{ij}),
\end{aligned}
$$

*where* $\mathrm{Sigmoid}(b) = \frac{1}{1+\exp(-b)}$. *Then,* $-\log p(\theta, z, x)$ *is convex and smooth with respect to* $(\theta, z)$.

*Proof.* It's immediate that $-\log p$ is convex with respect to $(\theta, z)$. We can show that this function is smooth, by verifying the three conditions of Theorem 27.

- $\nabla^2_{\theta\theta^\top} \log p(\theta, z, x) = -\Sigma^{-1} - \Delta^{-1}$ is constant.

- $\nabla^2_{\theta z_i^\top} \log p(\theta, z_i, x_i) = \Delta^{-1}$ is also constant.

- $p(z_i|\theta)$ is a Gaussian with constant covariance, so is smooth with a constant independent of $\theta$. And we can write that $\log p(x_{ij}|z_i) = \log\left(1 + \exp\left(-x_{ij}z_i^\top a_{ij}\right)\right)$. This function is also smooth over $z_i$ with a constant that doesn't depend on $\theta$. Thus $\log p(z_i|\theta) + \sum_j \log p(x_{ij}|z_i)$ is smooth over $z_i$ with a smoothness constant that is independent of $\theta$.

$\square$

## 8 Estimators and variance bounds

We remember that unless specified otherwise, we consider that $w = (m, C) \in \mathbb{R}^d \times V$, where $V$ is a vector subspace of $\mathcal{M}^d$.

### 8.1 Variance bounds for the estimators

**Theorem 1.** *Let $T_w(u) = Cu + m$ for $w = (m, C)$. Let $\phi : \mathbb{R}^d \to \mathbb{R}$ be $M$-smooth, suppose that $\phi$ is stationary at $\bar{m}$, and define $\bar{w} = (\bar{m}, 0)$. Then*

$$\mathbb{E}_{u \sim \mathcal{N}(0,I)} \|\nabla_w \phi(T_w(\mathsf{u}))\|_2^2 \leq (d+1)M^2 \|m - \bar{m}\|_2^2 + (d+3)M^2 \|C\|_F^2 \leq (d+3)M^2 \|w - \bar{w}\|_2^2.$$

*Furthermore, the first inequality cannot be improved.*

*Proof.* This bound is proven by [20, Thm. 3] in the case that $V = \mathcal{M}^d$. We conclude that this bound holds for every subspace $V$ of $\mathcal{M}^d$ by using the inequality in Lemma 17.2. $\square$

**Theorem 2.** *Suppose that $\log p(\cdot, x)$ is $M$-smooth and has a maximum (or stationary point) at $\bar{m}$, and define $\bar{w} = (\bar{m}, 0)$. Then, for every $w$ and every solution $w^*$ of the VI problem,*

$$\begin{aligned} \mathbb{E} \|\mathsf{g}_{\text{energy}}(\mathsf{u})\|_2^2 &\leq (d+3)M^2 \|w - \bar{w}\|_2^2 \\ &\leq 2(d+3)M^2 \|w - w^*\|_2^2 + 2(d+3)M^2 \|w^* - \bar{w}\|_2^2. \end{aligned} \tag{6}$$

*Proof.* It is a direct consequence of Theorem 1 and Young's inequality

$$\begin{aligned} \mathbb{E} \|\mathsf{g}_{\text{energy}}\|_2^2 &\leq (d+3)M^2 \|w - \bar{w}\|_2^2 \\ &= (d+3)M^2 \|w - w^* + w^* - \bar{w}\|_2^2 \\ &\leq 2(d+3)M^2 \|w - w^*\|_2^2 + 2(d+3)M^2 \|w^* - \bar{w}\|_2^2. \end{aligned}$$

$\square$

**Theorem 3.** *Suppose that $\log p(\cdot, x)$ is $M$-smooth, that it is maximal at $\bar{m}$, and define $\bar{w} = (\bar{m}, 0)$. Then, for every $L > 0$, for every $w \in \mathcal{W}_L$ and every solution $w^*$ of the VI problem,*

$$\begin{aligned} \mathbb{E} \|\mathsf{g}_{\text{ent}}(\mathsf{u})\|_2^2 &\leq 2(d+3)M^2 \|w - \bar{w}\|_2^2 + 2dL \\ &\leq 4(d+3)M^2 \|w - w^*\|^2 + 4(d+3)M^2 \|w^* - \bar{w}\|^2 + 2dL. \end{aligned} \tag{8}$$

*Proof.* The difference between $\mathsf{g}_{\text{ent}}$ and $\mathsf{g}_{\text{energy}}$ is the addition of the constant vector $\nabla h(w)$. Thus, we have that

$$\begin{aligned} \mathbb{E} \|\mathsf{g}_{\text{ent}}\|_2^2 &= \mathbb{E} \|\mathsf{g}_{\text{energy}} + \nabla h(w)\|_2^2 \\ &\leq 2\mathbb{E} \|\mathsf{g}_{\text{energy}}\|_2^2 + 2\mathbb{E} \|\nabla h(w)\|_2^2 \\ &\overset{(6)}{\leq} 2(d+3)M^2 \|w - \bar{w}\|_2^2 + 2 \|\nabla h(w)\|_2^2. \end{aligned}$$

It remains to bound the final term. We can do this using the closed-form expression for the entropy gradient (see Lemma 18.2), plus the assumption that $w \in \mathcal{W}_L \subset \mathcal{W}$, to see that

$$
\begin{aligned}
\|\nabla h(w)\|_2^2 &= \|\operatorname{proj}_V(C^{-\top})\|_F^2 \\
&\leq \|C^{-\top}\|_F^2 \\
&= \sum_i \sigma_i(C^{-\top})^2 \\
&= \sum_i \sigma_i(C)^{-2} \\
&\leq \sum_i \left(\frac{1}{\sqrt{L}}\right)^{-2} \\
&= dL.
\end{aligned}
$$

We conclude with Young's inequality

$$
\begin{aligned}
\mathbb{E}\|\mathsf{g}_{\mathrm{ent}}\|_2^2 &\leq 2(d+3)M^2\|w-\bar{w}\|_2^2 + dL \\
&= 2(d+3)M^2\|w-w^*+w^*-\bar{w}\|_2^2 + dL \\
&\leq 4(d+3)M^2\|w-w^*\|_2^2 + 4(d+3)M^2\|w^*-\bar{w}\|_2^2 + dL.
\end{aligned}
$$

$\square$

For the next Theorem, we will need the follwing technical Lemma :

**Lemma 29.** *Let* $\mathsf{u} \sim \mathcal{N}(0, I)$, $A \in \mathcal{M}^d$ *and* $b \in \mathbb{R}^d$. *Then*

$$
\mathbb{E}\|A\mathsf{u}+b\|^2(1+\|\mathsf{u}\|^2) = (d+1)\|b\|^2 + (d+3)\|A\|_F^2.
$$

*Proof.* Develop the squares to write

$$
\mathbb{E}\|A\mathsf{u}+b\|^2(1+\|\mathsf{u}\|^2) = \mathbb{E}_{\mathsf{u}}\|A\mathsf{u}\|^2 + 2\langle A^\top b, \mathsf{u}\rangle + \|b\|^2 + \|A\mathsf{u}\|^2\|\mathsf{u}\|^2 + 2\langle A^\top b, \mathsf{u}\|\mathsf{u}\|^2\rangle + \|b\|^2\|\mathsf{u}\|^2.
$$

Since $\mathbb{E}\mathsf{u} = 0$, we immediately see that $\mathbb{E}\langle A^\top b, \mathsf{u}\rangle = 0$. Because of symmetry, the third-order moment $\mathbb{E}\mathsf{u}\|\mathsf{u}\|^2$ is also zero, so we obtain $\mathbb{E}\langle A^\top b, \mathsf{u}\|\mathsf{u}\|^2\rangle = 0$. We also have the second-order moment $\mathbb{E}\|\mathsf{u}\|^2 = d$, which means that $\mathbb{E}\|b\|^2\|\mathsf{u}\|^2 = d\|b\|^2$. For the other terms, we use the identities

$$
\mathbb{E}\mathsf{u}\mathsf{u}^\top = I, \quad \mathbb{E}\mathsf{u}\mathsf{u}^\top\mathsf{u}\mathsf{u}^\top = (d+2)I,
$$

from [20, Lemma 9] It allows us to write

$$
\begin{aligned}
\mathbb{E}\|A\mathsf{u}\|^2 &= \mathbb{E}\operatorname{tr}(A^\top A\mathsf{u}\mathsf{u}^\top) = \operatorname{tr}(A^\top A) = \|A\|_F^2, \\
\mathbb{E}\|A\mathsf{u}\|^2\|\mathsf{u}\|^2 &= \operatorname{tr}(A^\top A\mathsf{u}\mathsf{u}^\top\mathsf{u}\mathsf{u}^\top) = (d+2)\|A\|_F^2.
\end{aligned}
$$

The conclusion follows after gathering all those identities. $\square$

**Theorem 4.** *Suppose that* $\log p(\cdot, x)$ *is $M$-smooth. Consider the residual* $r(z) := \log p(z, x) - \log q_{w^*}(z)$ *for any solution* $w^*$ *of the VI problem, assume that it has a stationary point* $\hat{m}$, *and define* $\hat{w} = (\hat{m}, 0)$. *Then* $r$ *is $K$-smooth for some* $K \in [0, 2M]$, *and for all* $w \in \mathcal{W}_M$,

$$
\begin{aligned}
\mathbb{E}\|\mathsf{g}_{\mathrm{STL}}\|_2^2 &\leq 8(d+3)M^2\|w-w^*\|_2^2 + 2(d+3)K^2\|w-\hat{w}\|_2^2 \quad (10) \\
&\leq 4(d+3)(K^2+2M^2)\|w-w^*\|_2^2 + 4(d+3)K^2\|w^*-\hat{w}\|_2^2.
\end{aligned}
$$

*Moreover, if* $p(\cdot|x)$ *is Gaussian then* $K = 0$.

*Proof.* Let $w \in \mathcal{W}_M$ be fixed. For the duration of the proof, we take $v$ to be a second copy of $w$ (held constant under differentiation with respect to $w$). Now, we can rearrange the estimator in Eq. 9 by making appear $q_{w^*}(T_w(\mathsf{u}))$, namely

$$
\mathsf{g}_{\mathrm{STL}}(\mathsf{u}) = \nabla_w \log \frac{q_v(T_w(\mathsf{u}))}{p(T_w(\mathsf{u}), x)} = \nabla_w \log \frac{q_v(T_w(\mathsf{u}))}{q_{w^*}(T_w(\mathsf{u}))} + \nabla_w \log \frac{q_{w^*}(T_w(\mathsf{u}))}{p(T_w(\mathsf{u}), x)}.
$$

Introducing $\phi(z) := \log q_v(z) - \log q_{w^*}(z)$, and recalling $r(z) = \log p(z, x) - \log q_{w^*}(z)$, we have

$$\mathsf{g}_{\mathrm{STL}} = \nabla_w \phi(T_w(\mathsf{u})) - \nabla_w r(T_w(\mathsf{u})).$$

We assume that $\log p(\cdot, x)$ is $M$-smooth, and that $V = \mathcal{M}^d, \mathcal{T}^d$ or $\mathcal{S}^d$, which implies that $w^* \in \mathcal{W}_M$ (see Lemma 20). This in turn guarantees that $\log q_{w^*}$ is also $M$-smooth (see Lemma 21). Thus, $r$ is the sum of two $M$-smooth functions, so it is $K$-smooth with $K \in [0, 2M]$. Using Young's inequality, together with Theorem 1 applied to $r$, we obtain

$$
\begin{aligned}
\mathbb{E}_{\mathsf{u}}\|\mathsf{g}_{\mathrm{STL}}(\mathsf{u})\|^2 &\leq 2\mathbb{E}_{\mathsf{u}}\|\nabla_w r(T_w(\mathsf{u}))\|^2 + 2\mathbb{E}_{\mathsf{u}}\|\nabla_w \phi(T_w(\mathsf{u}))\|^2 \\
&\leq 2(d+3)K^2\|w - \hat{w}\|^2 + 2\mathbb{E}_{\mathsf{u}}\|\nabla_w \phi(T_w(\mathsf{u}))\|^2,
\end{aligned}
\tag{19}
$$

where $\hat{w} = (\hat{m}, 0)$, with $\hat{m}$ being a stationary point of $r$. Now it remains to control the last term of inequality (19). Apply the chain rule onto $\phi(T_w(\mathsf{u}))$ to write (with the help of Lemma 37)

$$\nabla_w \phi(T_w(\mathsf{u})) = \left(\nabla_z \phi(T_w(\mathsf{u})), \mathrm{proj}_V(\nabla_z \phi(T_w(\mathsf{u}))\mathsf{u}^\top)\right).$$

This gives us directly that

$$
\begin{aligned}
\|\nabla_w \phi(T_w(\mathsf{u}))\|^2 &= \|\nabla_z \phi(T_w(\mathsf{u}))\|^2 + \|\mathrm{proj}_V(\nabla_z \phi(T_w(\mathsf{u}))\mathsf{u}^\top)\|_F^2 \\
&\leq \|\nabla_z \phi(T_w(\mathsf{u}))\|^2 + \|\nabla_z \phi(T_w(\mathsf{u}))\mathsf{u}^\top\|_F^2 \\
&= \|\nabla_z \phi(T_w(\mathsf{u}))\|^2 \left(1 + \|\mathsf{u}\|^2\right),
\end{aligned}
\tag{20}
$$

where we used that $\|vu^\top\|_F^2 = \|v\|^2\|u\|^2$ for any vectors $v$ and $u$. Computing $\nabla_z \phi$ amounts to computing the gradient of the Gaussian density $q_w$. From its definition (see Eq. 17) we have:

$$\nabla_z \log q_w(z) = -(CC^\top)^{-1}(z - m) = -\Sigma^{-1}(z - m).$$

Recalling the definition of $\phi = \log q_v - \log q_{w^*}$, and writing $v = (m, C)$, $w^* = (m_*, C_*)$, $\Sigma = CC^\top$, $\Sigma_* = C_* C_*^\top$, we obtain

$$\nabla_z \phi(z) = \left(\Sigma_*^{-1} - \Sigma^{-1}\right) z + \Sigma^{-1} m - \Sigma_*^{-1} m_*.$$

In other words, we have

$$\nabla_z \phi(T_w(\mathsf{u})) = \left(\Sigma_*^{-1} - \Sigma^{-1}\right)(C\mathsf{u} + m) + \Sigma^{-1} m - \Sigma_*^{-1} m_* = A\mathsf{u} + b, \tag{21}$$

where $A := (\Sigma_*^{-1} - \Sigma^{-1})C$ and $b := \Sigma_*^{-1} m - \Sigma_*^{-1} m_*$. We can now combine (20) and (21) and use Lemma 29 to write

$$
\begin{aligned}
\mathbb{E}_{\mathsf{u}}\|\nabla_w \phi(T_w(\mathsf{u}))\|^2 &\leq \mathbb{E}_{\mathsf{u}}\|A\mathsf{u} + b\|^2(1 + \|\mathsf{u}\|^2) \\
&= (d+1)\|b\|^2 + (d+3)\|A\|_F^2 \\
&\leq (d+3)\left(\|b\|^2 + \|A\|_F^2\right).
\end{aligned}
\tag{22}
$$

Let us now introduce the function $\kappa : \mathbb{R}^d \times \mathcal{M}^d \to \mathbb{R} \cup \{+\infty\}$, defined by $\kappa(w) := \mathbb{E}_{\mathsf{z} \sim q_w} \log q_w(\mathsf{z}) - \log q_{w^*}(\mathsf{z})$ if $C$ is invertible, $+\infty$ otherwise. This function is nothing but the Kullback-Liebler divergence between the two Gaussian distributions $q_w$ and $q_{w^*}$, and can be computed in closed-form on its domain:

$$\kappa(w) = \frac{1}{2}\left(\log\det(\Sigma_*) - \log\det(\Sigma) - d + \mathrm{tr}(\Sigma_*^{-1}\Sigma) + \langle \Sigma_*^{-1}(m_* - m), (m_* - m)\rangle\right),$$

where we note as before $w = (m, C)$, $w^* = (m_*, C_*)$, $\Sigma = CC^\top$, $\Sigma_* = C_* C_*^\top$. This allows us to compute its gradient at $w \in \mathcal{W}_M$:

$$
\begin{aligned}
\nabla_m \kappa(w) &= \Sigma_*^{-1}(m_* - m) \\
\nabla_C \kappa(w) &= \frac{-1}{2}\nabla_C \log\det(CC^\top) + \frac{1}{2}\nabla_C \mathrm{tr}(\Sigma_*^{-1}CC^\top) \\
&= (\Sigma_*^{-1} - \Sigma^{-1})C.
\end{aligned}
$$

We see that $\nabla_m \kappa(w) = b$, and $\nabla_C \kappa(w) = A$, which means that $\|b\|^2 + \|A\|_F^2 = \|\nabla_w \kappa(w)\|^2$. From (22) and the fact that $\nabla_w \kappa(w^*) = 0$ (because $w^*$ is a minimizer of $\kappa$), we deduce that

$$\mathbb{E}_{\mathsf{u}}\|\nabla_w \phi(T_w(\mathsf{u}))\|^2 \leq (d+3)\|\nabla_w \kappa(w) - \nabla_w \kappa(w^*)\|^2.$$

Now we remind that $w^* \in \mathcal{W}_M$ which implies that $\log q_{w^*}$ is $M$-smooth. This means that $\mathbb{E}_{\mathbf{z} \sim q_w} \log q_{w^*}$ is $M$-smooth as well, according to Domke [13, Theorem 1]. On the other hand, we know (see the proof of Lemma 16) that $\mathbb{E}_{\mathbf{z} \sim q_w} \log q_{w^*} = h(w)$ which is $M$-smooth on $\mathcal{W}_M$ (see Lemma 18.3). All this implies that $\kappa$ is $2M$-smooth on $\mathcal{W}_M$, from which we conclude

$$\mathbb{E}_{\mathbf{u}} \|\nabla_w \phi(T_w(\mathbf{u}))\|^2 \leq (d+3)\|\nabla_w \kappa(w) - \nabla_w \kappa(w^*)\|^2 \leq (d+3)4M^2\|w - w^*\|^2.$$

The Theorem's main inequality (10) follows after plugging the above inequality into (19). The second inequality follows after using Young's inequality

$$
\begin{aligned}
\mathbb{E} \|\mathbf{g}_{\mathrm{STL}}\|_2^2 &\leq 2(d+3)K^2 \|w - \hat{w}\|_2^2 + 8(d+3)M^2 \|w - w^*\|_2^2 \\
&= 2(d+3)K^2 \|w - w^* + w^* - \hat{w}\|_2^2 + 8(d+3)M^2 \|w - w^*\|_2^2 \\
&\leq 4(d+3)K^2 \|w - w^*\|_2^2 + 4(d+3)K^2 \|w^* - \hat{w}\|_2^2 + 8(d+3)M^2 \|w - w^*\|_2^2 \\
&= 4(d+3)\left(K^2 + 2M^2\right) \|w - w^*\|_2^2 + 4(d+3)K^2 \|w^* - \hat{w}\|_2^2.
\end{aligned}
$$

To conclude the proof, it remains to check that $K = 0$ whenever $p(\cdot|x)$ is Gaussian. In that case, because our main objective function $f(w)$ is the divergence between $q_w$ and $p(\cdot|x)$, it is clear that $f$ is minimized whenever $q_w = p(\cdot|x)$. Therefore, without loss of generality, we can assume that $p(\cdot|x) = q_{w^*}$. This implies that the residual $r$ is constant ($r(z) = \log p(x)$), and so is 0-smooth.

$\square$

## 8.2 Quadratically bounded estimators

**Theorem 30.** *The estimators obey $\mathbb{E} \|\mathbf{g}\|_2^2 \leq a \|w - w^*\|^2 + b$ with the the following constants:*

| *Estimator* | $a$ | $b$ | *valid $w$* |
|---|---|---|---|
| $\mathbf{g}_{\mathrm{energy}}$ | $2(d+3)M^2$ | $2(d+3)M^2 \|w^* - \bar{w}\|_2^2$ | *any* |
| $\mathbf{g}_{\mathrm{ent}}$ | $4(d+3)M^2$ | $4(d+3)M^2 \|w^* - \bar{w}\|_2^2 + dL$ | $w \in \mathcal{W}_L$ |
| $\mathbf{g}_{\mathrm{STL}}$ | $4(d+3)\left(K^2 + 2M^2\right)$ | $4(d+3)K^2 \|w^* - \hat{w}\|_2^2$ | $w \in \mathcal{W}_M$ |

*Proof.* Combine the results of Theorems 2, 3 and 4.

$\square$

# 9 Optimization proofs

This section contains all the proofs for the Theorems stated in Section 3.

## 9.1 Anytime Convergence Theorem for Strongly Convex

**Theorem 31** ( Theorem 3.2 in [53] ). *Suppose we are given a sequence of $w^t$ iterates such that for a step size $\gamma_t \leq \frac{1}{2\mathcal{L}}$ and given constants $\mu > 0$ and $\sigma^2 > 0$ we have that*

$$\mathbb{E}\left[\|w^{t+1} - w^*\|^2\right] \leq (1 - \gamma_t \mu)\mathbb{E}\left[\|w^t - w^*\|^2\right] + 2\sigma^2 \gamma_t^2. \tag{23}$$

*By switching to a decaying stepsize according to*

$$
\gamma_t = \begin{cases}
\dfrac{1}{2\mathcal{L}} & \text{for } t < t^* \\[2ex]
\dfrac{1}{\mu}\dfrac{2t+1}{(t+1)^2} & \text{for } t \geq t^*
\end{cases}
$$

*where $t^* = 4\lfloor \mathcal{L}/\mu \rfloor$, we have that*

$$\mathbb{E}\left[\|w^{T+1} - w^*\|^2\right] \leq \frac{16\lfloor \mathcal{L}/\mu \rfloor^2}{(T+1)^2}\|w^0 - w^*\|^2 + \frac{\sigma^2}{\mu^2}\frac{8}{T+1}. \tag{24}$$

*Proof.* We divide the proof for steps $t$ that are great or smaller than $t^*$. For $t \leq t^*$, we have by unrolling (23) starting from $t^*, \ldots, 1$, with $\gamma_t \equiv \gamma$ gives

$$\mathbb{E}\left[\|w^{t^*} - w^*\|^2\right] \leq (1 - \gamma\mu)^{t^*}\|w^0 - w^*\|^2 + \frac{2\sigma^2\gamma}{\mu}. \tag{25}$$

Now consider $t \geq t^*$ for which we have that $\gamma_t = \frac{1}{\mu}\frac{2t+1}{(t+1)^2}$ which when inserted into (23) gives

$$\mathbb{E}\left[\|w^{t+1} - w^*\|^2\right] \leq \frac{t^2}{(t+1)^2}\mathbb{E}\left[\|w^t - w^*\|^2\right] + \sigma^2\frac{2}{\mu^2}\frac{(2t+1)^2}{(t+1)^4}.$$

Multiplying through by $(t+1)^2$ and re-arranging gives

$$(t+1)^2\mathbb{E}\left[\|w^{t+1} - w^*\|^2\right] \leq t^2\mathbb{E}\left[\|w^t - w^*\|^2\right] + \sigma^2\frac{2}{\mu^2}\frac{(2t+1)^2}{(t+1)^2}$$

$$\leq t^2\mathbb{E}\left[\|w^t - w^*\|^2\right] + \frac{8\sigma^2}{\mu^2},$$

where we used that $\frac{2t+1}{t+1} \leq 2$. Summing up over $t = t^*, \ldots, T$ and using telescopic cancellation gives

$$(T+1)^2\mathbb{E}\left[\|w^{T+1} - w^*\|^2\right] \leq (t^*)^2\mathbb{E}\left[\|w^{t^*} - w^*\|^2\right] + (T - t^*)\frac{8\sigma^2}{\mu^2}.$$

Now using that for $t \leq t^*$ we have that $\gamma = \frac{1}{\mathcal{L}}$ and (25) holds, thus

$$(T+1)^2\mathbb{E}\left[\|w^{T+1} - w^*\|^2\right] \leq (t^*)^2\left((1 - \gamma\mu)^{t^*}\|w^0 - w^*\|^2 + 2\sigma^2\frac{\gamma}{\mu}\right) + (T - t^*)\frac{8\sigma^2}{\mu^2}$$

$$\leq (t^*)^2\left(\|w^0 - w^*\|^2 + 2\sigma^2\frac{\gamma}{\mu}\right) + (T - t^*)\frac{8\sigma^2}{\mu^2},$$

where we used that $(1 - \gamma\mu) \leq 1$. Substituting $\gamma = \frac{1}{2\mathcal{L}}$, $t^* = 4\lfloor\mathcal{L}/\mu\rfloor$ and multiplying by $(T+1)^2$ gives

$$\mathbb{E}\left[\|w^{T+1} - w^*\|^2\right] \leq \frac{16\lfloor\mathcal{L}/\mu\rfloor^2}{(T+1)^2}\|w^0 - w^*\|^2 + \frac{\sigma^2}{\mu^2(T+1)^2}\left(8(T - t^*) + (t^*)^2\frac{\mu}{\mathcal{L}}\right)$$

$$\leq \frac{16\lfloor\mathcal{L}/\mu\rfloor^2}{(T+1)^2}\|w^0 - w^*\|^2 + \frac{\sigma^2}{\mu^2}\frac{8(T - 2t^*)}{(T+1)^2}$$

$$\leq \frac{16\lfloor\mathcal{L}/\mu\rfloor^2}{(T+1)^2}\|w^0 - w^*\|^2 + \frac{\sigma^2}{\mu^2}\frac{8}{T+1}$$

which concludes the proof of (24). $\qquad\square$

## 9.2 Complexity Lemma for Convex

**Lemma 32.** *Suppose we are given a sequence $\bar{w}^T$ and constants $a, b, B > 0$ such that*

$$\mathbb{E}\left[f(\bar{w}^T)\right] - \inf f \leq \gamma\left(\frac{a\|w^0 - w^*\|^2}{1 - \theta^T} + b\right)$$

*holds for*

$$\theta := \frac{1}{1 + B\gamma^2}.$$

*If $\gamma = \frac{A}{\sqrt{T}}$ for $A \geq \sqrt{\frac{2}{B}}$ and $T \geq 2$, then*

$$\mathbb{E}\left[f(\bar{w}^T)\right] - \inf f \leq \frac{A}{\sqrt{T}}\left(2a\|w^0 - w^*\|^2 + b\right).$$

*Thus, $T = \mathcal{O}(\epsilon^{-2})$ iterations are sufficient to guarantee that $\mathbb{E}\left[f(\bar{w}^T)\right] - \inf f \leq \epsilon$.*

*Proof.* First we show that $\frac{1}{1-\theta^T} \leq 2$ is equivalent to

$$T \geq \frac{\log 2}{\log(1 + BA^2/T)}. \tag{26}$$

Indeed this follows since

$$\frac{1}{1 - \theta^T} \leq 2 \quad \Leftrightarrow$$

$$\theta^T \leq \frac{1}{2} \quad \Leftrightarrow$$

$$T \geq \frac{\log 2}{\log 1/\theta} \quad \Leftrightarrow$$

$$T \geq \frac{\log 2}{\log(1 + B\gamma^2)} \quad \Leftrightarrow$$

$$T \geq \frac{\log 2}{\log(1 + BA^2/T)}.$$

Now note that $\log(1+x) \geq \frac{x}{1+x}$ for $x \geq 0$, which implies that $\frac{1}{\log(1+x)} \leq 1 + \frac{1}{x}$ for $x \geq 0$. Applying this gives that

$$\frac{1}{\log(1 + BA^2/T)} \leq 1 + \frac{T}{BA^2}.$$

So to guarantee $\frac{1}{1-\theta^T} \leq 2$, from (26) it is sufficient to enforce that

$$T \quad \geq \quad 1 + \frac{T}{BA^2}$$
$$\geq \quad \log(2)\left(1 + \frac{T}{BA^2}\right).$$

Assuming $A \geq \sqrt{2/B}$, this last condition holds if

$$T \geq 1 + \frac{1}{BA^2 - 1}. \tag{27}$$

Since we also impose that $BA^2 \geq 2$ we have that $\frac{1}{BA^2-1} \leq 1$ thus for (27) to hold it suffices that $T \geq 2$. Substituting $\gamma = \frac{A}{\sqrt{T}}$ and the relaxation $\frac{1}{1-\theta^T} \leq 2$ into the original bound gives the claimed result. $\square$

## 9.3 Proximal gradient descent with strong convexity and smoothness

We start by recalling some standard properties of the proximal operator.

**Lemma 33.** *Let $\gamma > 0$ and $w^* \in \operatorname{argmin}_w l(w) + h(w)$. Assume $h(w)$ is convex and that $l(w)$ is continuously differentiable at $w^*$. Then*

1. *For all $w, w'$ and $\gamma > 0$, $\left\|\operatorname{prox}_{\gamma h}(w) - \operatorname{prox}_{\gamma h}(w')\right\|_2 \leq \|w - w'\|_2$.*

2. *$w^* = \operatorname{prox}_{\gamma h}(w^* - \gamma \nabla l(w^*))$*

3. *$v = \operatorname{prox}_{\gamma h}(w) \Leftrightarrow \frac{w-v}{\gamma} \in \partial h(v)$*

**Theorem 7.** *Let $l$ be a $\mu$-strongly convex and $M$-smooth function, and let $\bar{w} = \operatorname{argmin}(l)$. Let $h$ be a proper closed convex function, and let $w^* = \operatorname{argmin}(l + h)$. Let $(w^t)_{t \in \mathbb{N}}$ be generated by the Prox-SGD algorithm, with a constant stepsize $\gamma \in \left(0, \min\{\frac{\mu}{2a}, \frac{1}{\mu}\}\right]$. Suppose that $g^t$ is a quadratically bounded estimator (Def. 5) for $\nabla l$ with parameters $(a, b, w^*)$. Then,*

$$\mathbb{E}\left\|w^{T+1} - w^*\right\|_2^2 \leq (1 - \gamma\mu)^T \left\|w^0 - w^*\right\|_2^2 + \frac{2\gamma}{\mu}\left(b + M^2 \left\|w^* - \bar{w}\right\|_2^2\right). \tag{11}$$

*Alternatively, if we use the decaying stepsize $\gamma_t = \min\left\{\dfrac{\mu}{2a}, \dfrac{1}{\mu}\dfrac{2t+1}{(t+1)^2}\right\}$, then*

$$\mathbb{E}\|w^T - w^*\|^2 \quad \leq \quad \frac{16\lfloor a/\mu^2\rfloor^2}{T^2}\|w^0 - w^*\|^2 + \frac{8}{\mu^2 T}\left(b + M^2\,\|w^* - \bar{w}\|_2^2\right). \tag{12}$$

*In both cases, $T = \mathcal{O}(\epsilon^{-1})$ iterations are sufficient to guarantee that $\mathbb{E}\|w^T - w^*\|^2 \leq \epsilon$.*

*Proof.* We start by using the non-expansiveness and fixed-point properties of the proximal operator to write

$$
\begin{aligned}
\left\|w^{t+1} - w^*\right\|_2^2 \quad &= \quad \left\|\text{prox}_{\gamma h}\left(w^t - \gamma g^t\right) - \text{prox}_{\gamma h}\left(w^* - \gamma \nabla l(w^*)\right)\right\|_2^2 \\
&\leq \quad \left\|w^t - w^* + \gamma\left(\nabla l(w^*) - g^t\right)\right\|_2^2 \\
&= \quad \left\|w^t - w^*\right\|_2^2 + \gamma^2\left\|\nabla l(w^*) - g^t\right\|_2^2 + 2\gamma\left\langle w^t - w^*,\ \nabla l(w^*) - g^t\right\rangle.
\end{aligned}
$$

Above the first line uses the fixed point property of the proximal operator that $w^* = \text{prox}_{\gamma h}(w^* - \gamma \nabla l(w))$, while the second line uses the fact that the proximal operator is non-expansive.

Now, we apply our assumption on the gradient estimator to write

$$
\begin{aligned}
\mathbb{E}\left[\left\|\nabla l(w^*) - g^t\right\|_2^2 \mid w^t\right] \quad &\leq \quad 2\,\mathbb{E}\left[\|\nabla l(w^*)\|_2^2 \mid w^t\right] + 2\,\mathbb{E}\left[\|g^t\|_2^2 \mid w^t\right] \\
&\leq \quad 2\|\nabla l(w^*)\|_2^2 + 2\left(a\left\|w^t - w^*\right\|_2^2 + b\right) \\
&\leq \quad 2M^2\left\|w^* - \bar{w}\right\|_2^2 + 2\left(a\left\|w^t - w^*\right\|_2^2 + b\right).
\end{aligned}
$$

For the last term, we have by strong convexity and the fact that $g^t$ is an unbiased estimator that

$$
\begin{aligned}
\mathbb{E}\left[\left\langle w^t - w^*,\ \nabla l(w^*) - g^t\right\rangle \mid w_t\right] \quad &= \quad -\left\langle w^* - w^t,\ \nabla l(w^*) - \nabla l(w^t)\right\rangle \\
&\leq \quad -\mu\left\|w^t - w^*\right\|_2^2,
\end{aligned}
$$

where the second line follows from the fact that $l$ is $\mu$ strongly convex.

Putting the pieces together, we get that

$$
\begin{aligned}
\mathbb{E}\left[\left\|w^{t+1} - w^*\right\|_2^2\right] \quad &\leq \quad \mathbb{E}\left\|w^t - w^*\right\|_2^2 + 2\gamma^2 M^2\left\|w^* - \bar{w}\right\|_2^2 + 2\gamma^2\,\mathbb{E}\left(a\left\|w^t - w^*\right\|_2^2 + b\right) - 2\gamma\mu\,\mathbb{E}\left\|w^t - w^*\right\| \\
&= \quad \left(1 - 2\gamma\mu + 2\gamma^2 a\right)\mathbb{E}\left\|w^t - w^*\right\|_2^2 + 2\gamma^2\left(b + M^2\left\|w^* - \bar{w}\right\|_2^2\right)
\end{aligned}
$$

Now the following conditions are equivalent:

$$
\begin{aligned}
1 - 2\gamma\mu + 2\gamma^2 a \quad &\leq \quad 1 - \gamma\mu \\
2\gamma\mu - 2\gamma^2 a \quad &\geq \quad \gamma\mu \\
2\mu - 2\gamma a \quad &\geq \quad \mu \\
2\mu - \mu \quad &\geq \quad 2\gamma a \\
\mu \quad &\geq \quad 2\gamma a \\
\gamma \quad &\leq \quad \frac{\mu}{2a}
\end{aligned}
$$

This means that

$$\mathbb{E}\left[\left\|w^{t+1} - w^*\right\|_2^2\right] \quad \leq \quad (1 - \gamma\mu)\,\mathbb{E}\left\|w^t - w^*\right\|_2^2 + 2\gamma^2\left(b + M^2\left\|w^* - \bar{w}\right\|_2^2\right).$$

Now since $\gamma \leq \frac{1}{\mu} \Leftrightarrow 1 - \gamma\mu \geq 0$ we can apply the above recursively, which gives that

$$\mathbb{E}\left[\left\|w^{t+1} - w^*\right\|_2^2\right] \quad \leq \quad (1 - \gamma\mu)^t\left\|w^0 - w^*\right\|_2^2 + 2\gamma^2\left(b + M^2\left\|w^* - \bar{w}\right\|_2^2\right)\sum_{k=1}^{t-1}(1 - \gamma\mu)^k.$$

But we can bound this geometric sum by

$$\sum_{k=1}^{t-1}(1 - \gamma\mu)^k = \frac{1 - (1 - \gamma\mu)^t}{\gamma\mu} \leq \frac{1}{\gamma\mu},$$

leading to the result that

$$\mathbb{E}\left[\left\|w^{t+1} - w^*\right\|_2^2\right] \leq (1 - \gamma\mu)^t \left\|w^0 - w^*\right\|_2^2 + \frac{2\gamma}{\mu}\left(b + M^2 \left\|w^* - \bar{w}\right\|_2^2\right).$$

To prove the anytime result, we can now apply Theorem 31. To do so, we need to map the notation we use here to the notation used in Theorem 31. The mapping we need is $\mathcal{L} = \frac{a}{\mu}$, and $\sigma^2 = \left(b + M^2 \left\|w^* - \bar{w}\right\|_2^2\right)$ which when inserting into Theorem 31 gives the result. $\qquad\square$

**Corollary 34.** *Under the conditions of Theorem 7, if $\gamma = \frac{A \log T}{T}$ for $\frac{1}{\mu} \leq A$, and $T$ is large enough that $\frac{T}{\log T} \geq \frac{2aA}{\mu}$,*

$$\mathbb{E}\left\|w^{T+1} - w^*\right\|_2^2 \leq \frac{1}{T}\left\|w^0 - w^*\right\|_2^2 + \frac{2A \log T}{\mu T}\left(b + M^2 \left\|w^* - \bar{w}\right\|_2^2\right) = \Omega\left(\frac{\log T}{T}\right).$$

*Proof.* The previous theorem can only be applied when $\gamma \leq \frac{\mu}{2a}$, i.e. when $\frac{A \log T}{T} \leq \frac{\mu}{2a}$ or $\frac{2aA}{\mu} \leq \frac{T}{\log T}$. Supposing that is true, observe that $(1 - x)^T \leq \exp(-xT)$, from which it follows that $(1 - \gamma\mu)^T \leq \exp(-\mu A \log T) = \frac{1}{T^{\mu A}} \leq \frac{1}{T}$. $\qquad\square$

### 9.4 Proximal gradient descent with convexity and smoothness

**Theorem 8.** *Let $l$ be a proper convex and $M$-smooth function. Let $h$ be a proper closed convex function, and let $w^* \in \operatorname{argmin}(l + h)$. Let $(w^t)_{t \in \mathbb{N}}$ be generated by the Prox-SGD algorithm, with a constant stepsize $\gamma \in (0, \frac{1}{M}]$. Suppose that $g^t$ is a quadratically bounded estimator (Def. 5) for $\nabla l$ with parameters $(a, b, w^*)$. Then,*

$$\mathbb{E}\left[f(\bar{w}^T) - \inf f\right] \leq \gamma\left(a\frac{\|w^0 - w^*\|^2}{(1 - \theta^T)} + b\right),$$

*where $\theta \overset{\text{def}}{=} \frac{1}{1 + 2a\gamma^2}$ and $\bar{w}^T \overset{\text{def}}{=} \frac{\sum_{t=1}^{T}\theta^{t+1}w^t}{\sum_{t=1}^{T}\theta^{t+1}}$. In particular, if $\gamma = \frac{1}{\sqrt{aT}}$, then*

$$\mathbb{E}\left[f(\bar{w}^T)\right] - \inf f \leq \frac{1}{\sqrt{aT}}\left(2a\|w^0 - w^*\|^2 + b\right) \quad \forall T \geq \max\left\{\frac{M^2}{a}, 2\right\}.$$

*Thus, $T = \mathcal{O}(\epsilon^{-2})$ iterations are sufficient to guarantee that $\mathbb{E}\left[f(\bar{w}^T) - \inf f\right] \leq \epsilon$.*

*Proof.* Let us start by looking at $\|w^{t+1} - w^*\|^2 - \|w^t - w^*\|^2$. Expanding the squares, we have that

$$\frac{1}{2\gamma_t}\|w^{t+1} - w^*\|^2 - \frac{1}{2\gamma_t}\|w^t - w^*\|^2 = \frac{-1}{2\gamma_t}\|w^{t+1} - w^t\|^2 - \langle\frac{w^t - w^{t+1}}{\gamma_t}, w^{t+1} - w^*\rangle.$$

Since $w^{t+1} = \operatorname{prox}_{\gamma_t h}(w^t - \gamma_t g^t)$, we know from Lemma 33 that

$$\frac{(w^t - \gamma_t g^t) - w^{t+1}}{\gamma_t} \in \partial h(w^{t+1})$$

and therefore that

$$\frac{w^t - w^{t+1}}{\gamma_t} \in g^t + \partial h(w^{t+1}),$$

where $\partial h(w)$ is the subdifferential of $h$. Consequently there exists a subgradient $b^{t+1} \in \partial h(w^{t+1})$ such that

$$\frac{1}{2\gamma_t}\|w^{t+1} - w^*\|^2 - \frac{1}{2\gamma_t}\|w^t - w^*\|^2 \qquad\qquad (28)$$

$$= \frac{-1}{2\gamma_t}\|w^{t+1} - w^t\|^2 - \langle g^t + b^{t+1}, w^{t+1} - w^*\rangle$$

$$= \frac{-1}{2\gamma_t}\|w^{t+1} - w^t\|^2 - \langle g^t - \nabla\ell(w^t), w^{t+1} - w^*\rangle - \langle\nabla\ell(w^t) + b^{t+1}, w^{t+1} - w^*\rangle.$$

We decompose the last term of (28) as

$$-\langle \nabla\ell(w^t)+b^{t+1}, w^{t+1}-w^*\rangle = -\langle b^{t+1}, w^{t+1}-w^*\rangle - \langle \nabla\ell(w^t), w^{t+1}-w^t\rangle + \langle \nabla\ell(w^t), w^*-w^t\rangle. \tag{29}$$

For the first term in the above we can use that $b^{t+1} \in \partial h(w^{t+1})$ is a subgradient together with the defining property of subgradient to write

$$-\langle b^{t+1}, w^{t+1}-w^*\rangle = \langle b^{t+1}, w^*-w^{t+1}\rangle \le h(w^*) - h(w^{t+1}). \tag{30}$$

On the second term we can use the fact that $l$ is $M$-smooth to write

$$-\langle \nabla\ell(w^t), w^{t+1}-w^t\rangle \le \frac{M}{2}\|w^{t+1}-w^t\|^2 + \ell(w^t) - \ell(w^{t+1}). \tag{31}$$

On the last term we can use the convexity of $f$ to write

$$\langle \nabla\ell(w^t), w^*-w^t\rangle \le \ell(w^*) - \ell(w^t). \tag{32}$$

By inserting (30), (31), (32) into (29) gives

$$-\langle \nabla\ell(w^t) + g^{t+1}, w^{t+1}-w^*\rangle \le h(w^*) + \ell(w^*) - h(w^{t+1}) - \ell(w^{t+1}) + \frac{M}{2}\|w^{t+1}-w^t\|^2$$

$$= f(w^*) - f(w^{t+1}) + \frac{M}{2}\|w^{t+1}-w^t\|^2.$$

Now using the above in (28), and our assumption that $\gamma_t M \le 1$, we obtain

$$\frac{1}{2\gamma_t}\|w^{t+1}-w^*\|^2 - \frac{1}{2\gamma_t}\|w^t-w^*\|^2$$

$$\le \frac{-1}{2\gamma_t}\|w^{t+1}-w^t\|^2 + \frac{L}{2}\|w^{t+1}-w^t\|^2 - (f(w^{t+1}) - \inf f) - \langle g^t - \nabla\ell(w^t), w^{t+1}-w^*\rangle$$

$$\le -(f(w^{t+1}) - \inf f) - \langle g^t - \nabla\ell(w^t), w^{t+1}-w^*\rangle. \tag{33}$$

We now have to control the last term of (33) in expectation. To shorten our notation we temporarily introduce the operators

$$T(w) \overset{\text{def}}{=} w - \gamma_t \nabla\ell(w),$$

$$\hat{T}(w) \overset{\text{def}}{=} w - \gamma_t g^t. \tag{34}$$

Notice in particular that $w^{t+1} = \text{prox}_{\gamma_t h}(\hat{T}(w^t))$. So the the last term of (33) can be decomposed as

$$-\langle g^t - \nabla\ell(w^t), w^{t+1}-w^*\rangle = -\langle g^t - \nabla\ell(w^t), \text{prox}_{\gamma_t h}(\hat{T}(w^t)) - \text{prox}_{\gamma_t h}(T(w^t))\rangle$$

$$-\langle g^t - \nabla\ell(w^t), \text{prox}_{\gamma_t h}(T(w^t)) - w^*\rangle. \tag{35}$$

We observe that the last term is, in expectation, is equal to zero. This is because $\text{prox}_{\gamma_t h}(T(w^t)) - w^*$ is deterministic when conditioned on $w^t$. Since we will later on take expectations, we drop this term now and keep on going. As for the first term, using the nonexpansiveness of the proximal operator (Lemma 33), we have that

$$-\langle g^t - \nabla\ell(w^t), \text{prox}_{\gamma_t h}(\hat{T}(w^t)) - \text{prox}_{\gamma_t h}(T(w^t))\rangle \le \|g^t - \nabla\ell(w^t)\|\|\hat{T}(w^t) - T(w^t)\|$$

$$= \gamma_t \|g^t - \nabla\ell(w^t)\|^2.$$

Using the above two bounds in (35) we have proved that (after taking expectation)

$$-\mathbb{E}\langle g^t - \nabla\ell(w^t), w^{t+1}-w^*\rangle \le \gamma_t \mathbb{E}\|g^t - \nabla\ell(w^t)\|^2 = \gamma_t \mathbb{V}\left[g^t\right].$$

Injecting the above inequality into (33) and multiplying through by $\gamma_t$, we obtain

$$\frac{1}{2}\mathbb{E}\|w^{t+1}-w^*\|^2 - \frac{1}{2}\mathbb{E}\left[\|w^t-w^*\|^2\right] \le -\gamma_t \mathbb{E}\left[f(w^{t+1}) - \inf f\right] + \gamma_t^2 \mathbb{V}\left[g^t\right]. \tag{36}$$

From now on we assume that the stepsize sequence is constant $\gamma_t \equiv \gamma$. Now, applying our assumption on the gradient estimator, we have

$$\mathbb{V}\left[g^t\right] \le \mathbb{E}\|g^t\|^2 \le a\,\mathbb{E}\left\|w^t-w^*\right\|^2 + b$$

Inject this inequality into (36), and reordering the terms gives

$$
\begin{aligned}
\gamma \mathbb{E}\left[f(w^{t+1}) - \inf f\right] &\leq \frac{1}{2}\mathbb{E}\left[\|w^t - w^*\|^2\right] - \frac{1}{2}\mathbb{E}\left[\|w^{t+1} - w^*\|^2\right] + \gamma^2\left(a\,\mathbb{E}\left[\|w^t - w^*\|^2\right] + b\right) \\
&= \frac{1}{2}\left(1 + 2\gamma^2 a\right)\mathbb{E}\left[\|w^t - w^*\|^2\right] - \frac{1}{2}\mathbb{E}\left[\|w^{t+1} - w^*\|^2\right] + \gamma^2 b \\
&= \frac{1}{2}C\,\mathbb{E}\left[\|w^t - w^*\|^2\right] - \frac{1}{2}\mathbb{E}\left[\|w^{t+1} - w^*\|^2\right] + \gamma^2 b \qquad (37)
\end{aligned}
$$

where we introduced the constant $C \overset{\text{def}}{=} 1 + 2a\gamma^2$. We will now do a weighted telescoping by introducing a sequence $\alpha_{t+1} > 0$ of weights. Multiplying the above inequality by some $\alpha_{t+1} > 0$, and summing over $0, \dots, T-1$ gives

$$
\gamma \sum_{t=0}^{T-1} \alpha_{t+1}\mathbb{E}\left[f(w^{t+1}) - \inf f\right] \leq \sum_{t=0}^{T-1} \alpha_{t+1}\left(C\frac{1}{2}\mathbb{E}\left[\|w^t - w^*\|^2\right] - \frac{1}{2}\mathbb{E}\left[\|w^{t+1} - w^*\|^2\right]\right) + \gamma^2 b \sum_{t=0}^{T-1} \alpha_{t+1}.
$$

To be able to use a telescopic sum in the right-hand term, we need that $\alpha_{t+1}C = \alpha_t$ (an idea we borrowed from [54] ). This holds by choosing $\alpha_t \overset{\text{def}}{=} \frac{1}{C^t}$, which allows us to write

$$
\begin{aligned}
\gamma \sum_{t=0}^{T-1} \alpha_{t+1}\mathbb{E}\left[f(w^{t+1}) - \inf f\right] &\leq \sum_{t=0}^{T-1}\left(\frac{\alpha_t}{2}\mathbb{E}\left[\|w^t - w^*\|^2\right] - \frac{\alpha_{t+1}}{2}\mathbb{E}\left[\|w^{t+1} - w^*\|^2\right]\right) + \gamma^2 b \sum_{t=0}^{T-1} \alpha_{t+1} \\
&= \frac{\alpha_0}{2}\|w^0 - w^*\|^2 - \frac{\alpha_T}{2}\mathbb{E}\left[\|w^T - w^*\|^2\right] + \gamma^2 b \sum_{t=0}^{T-1} \alpha_{t+1} \\
&\leq \frac{\alpha_0}{2}\|w^0 - w^*\|^2 + \gamma^2 b \sum_{t=0}^{T-1} \alpha_{t+1}.
\end{aligned}
$$

Let us now introduce $\bar{w}^T \overset{\text{def}}{=} \frac{\sum_{t=0}^{T-1} \alpha_{t+1} w^{t+1}}{\sum_{t=0}^{T-1} \alpha_{t+1}}$, using that $\alpha_0 = 1$, and by using the convexity of $f$ with Jensen's inequality, we obtain

$$
\begin{aligned}
\mathbb{E}\left[f(\bar{w}^T)\right] - \inf f &= \mathbb{E}\left[\frac{\sum_{t=0}^{T-1} \alpha_{t+t} f(w^{t+1})}{\sum_{t=0}^{T-1} \alpha_{t+1}}\right] - \inf f \\
&\leq \frac{\sum_{t=0}^{T-1} \alpha_{t+t}\,\mathbb{E}\left[f(w^{t+1})\right]}{\sum_{t=0}^{T-1} \alpha_{t+1}} - \inf f \\
&= \frac{\sum_{t=0}^{T-1} \alpha_{t+t}\,\mathbb{E}\left[f(w^{t+1}) - \inf f\right]}{\sum_{t=0}^{T-1} \alpha_{t+1}} \\
&= \frac{\sum_{t=0}^{T-1} \alpha_{t+t}\,\mathbb{E}\left[f(w^{t+1}) - \inf f\right]}{\sum_{t=0}^{T-1} \alpha_{t+1}} \\
&\leq \frac{\frac{1}{2\gamma}\|w^0 - w^*\|^2 + \gamma b \sum_{t=0}^{T-1} \alpha_{t+1}}{\sum_{t=0}^{T-1} \alpha_{t+1}} \\
&= \frac{1}{2\gamma \sum_{t=0}^{T-1} \alpha_{t+1}}\|w^0 - w^*\|^2 + \gamma b \\
&= \frac{1}{2\gamma \sum_{t=0}^{T-1} \alpha_{t+1}}\|w^0 - w^*\|^2 + \gamma b \\
&= \frac{1}{2\gamma \sum_{t=0}^{T-1} \alpha_{t+1}}\|w^0 - w^*\|^2 + \gamma b.
\end{aligned}
$$

To finish the proof, it remains to compute the geometric series $\sum_{t=0}^{T-1} \alpha_{t+1}$ which is given by

$$
\sum_{t=0}^{T-1} \alpha_{t+1} = \frac{1}{C}\sum_{t=0}^{T-1}\frac{1}{C^t} = \frac{1}{C}\frac{1 - 1/C^T}{1 - 1/C} = \frac{1 - 1/C^T}{C - 1}.
$$

Using the above and substituting for $C = 1 + 2a\gamma^2$ and $\theta = 1/C$ gives

$$
\begin{aligned}
\mathbb{E}\left[f(\bar{w}^T)\right] - \inf f &\leq \frac{C-1}{2\gamma\left(1 - 1/C^T\right)}\|w^0 - w^*\|^2 + \gamma b \\
&= \frac{a\gamma}{(1 - \theta^T)}\|w^0 - w^*\|^2 + \gamma b
\end{aligned}
$$

Finally by applying Lemma 32 with where $B = 2a$, $A = \sqrt{\frac{2}{B}} = \frac{1}{\sqrt{a}}$ and thus $\gamma = \frac{A}{\sqrt{T}} = \frac{1}{\sqrt{aT}}$. Note that with this choice of $\gamma$ we have that

$$
\gamma = \frac{1}{\sqrt{aT}} \leq \frac{1}{M} \;\Leftrightarrow\; T \geq \frac{M^2}{a}.
$$

Thus for $T \geq 2$ the result of Lemma 32 gives

$$
\mathbb{E}\left[f(\bar{w}^T)\right] - \inf f \;\leq\; \frac{1}{\sqrt{a}\sqrt{T}}\left(2a\|w^0 - w^*\|^2 + b\right) \;=\; \Omega\left(\frac{1}{\sqrt{T}}\right).
$$

$\square$

## 9.5 Projected gradient descent with strong convexity

**Theorem 10.** *Let $\mathcal{W}$ be a nonempty closed convex set. Let $f$ be a $\mu$-strongly convex function, differentiable on $\mathcal{W}$. Let $w^* = \operatorname{argmin}_{\mathcal{W}}(f)$. Let $(w^t)_{t \in \mathbb{N}}$ be generated by the Proj-SGD algorithm, with a constant stepsize $\gamma \in \left(0, \min\{\frac{\mu}{2a}, \frac{2}{\mu}\}\right]$. Suppose that $g^t$ is a quadratically bounded estimator (Def. 5) for $\nabla f$ with parameters $(a, b, w^*)$. Then,*

$$
\mathbb{E}\left\|w^T - w^*\right\|^2 \leq \left(1 - \frac{\mu\gamma}{2}\right)^T \|w^0 - w^*\|^2 + \frac{2\gamma b}{\mu}. \tag{13}
$$

*Alternatively, if we use the decaying stepsize $\gamma_t = \min\left\{\dfrac{\mu}{2a}, \dfrac{2}{\mu}\dfrac{2t+1}{(t+1)^2}\right\}$, then*

$$
\mathbb{E}\left[\|w^T - w^*\|^2\right] \;\leq\; \frac{32a}{\mu^2 T^2}\|w^0 - w^*\|^2 + \frac{16b}{\mu^2 T}. \tag{14}
$$

*In both cases, $T = \mathcal{O}(\epsilon^{-1})$ iterations are sufficient to guarantee that $\mathbb{E}\|w^T - w^*\|^2 \leq \epsilon$.*

*Proof.* Plugging in one step of Proj-SGD we have that

$$
\begin{aligned}
\left\|w^{t+1} - w^*\right\|^2 &= \left\|\operatorname{proj}_{\mathcal{W}}(w^t - \gamma_t g^t) - \operatorname{proj}_{\mathcal{W}} w^*\right\|^2 \\
&\leq \left\|w^t - w^* - \gamma_t g^t\right\|^2,
\end{aligned}
$$

where we used that projections are non-expansive. Taking expectation conditioned on $w^t$, and expanding squares we have that

$$
\begin{aligned}
\mathbb{E}\left[\left\|w^{t+1} - w^*\right\|^2 \mid w^t\right] &\leq \left\|w^t - w^*\right\|^2 - 2\gamma_t\left\langle\nabla f(w^t), w^t - w^*\right\rangle + \gamma_t^2\, \mathbb{E}\left[\left\|g^t\right\|^2 \mid w^t\right] \\
&\leq (1 - \mu\gamma_t)\left\|w^t - w^*\right\|^2 - 2\gamma_t(f(w^t) - f(w^*)) + a\gamma_t^2\|w^t - w^*\|^2 + b\gamma_t^2,
\end{aligned}
$$

where we used $\mathbb{E}\left[g^t\right] = \nabla f(w^t)$ and definition 5, and that $f$ is $\mu$–strongly convex. Taking full expectation and re-arranging gives

$$
2\gamma_t\, \mathbb{E}[f(w^t) - f(w^*)] \leq (1 - \gamma_t(\mu - a\gamma_t))\, \mathbb{E}\left\|w^t - w^*\right\|^2 - \mathbb{E}\left\|w^{t+1} - w^*\right\|^2 + b\gamma_t^2.
$$

Now using that $\gamma_t \leq \frac{\mu}{2a}$ we have that

$$
1 - \gamma_t(\mu - a\gamma_t) \leq 1 - \frac{\gamma_t \mu}{2}
$$

and thus

$$
2\gamma_t\, \mathbb{E}[f(w^t) - f(w^*)] \leq \left(1 - \frac{\mu\gamma_t}{2}\right)\mathbb{E}\left\|w^t - w^*\right\|^2 - \mathbb{E}\left\|w^{t+1} - w^*\right\|^2 + b\gamma_t^2. \tag{38}
$$

Re-arranging we have that

$$\mathbb{E} \left\| w^{t+1} - w^* \right\|^2 \leq \left(1 - \frac{\mu\gamma_t}{2}\right) \mathbb{E} \left\| w^t - w^* \right\|^2 - 2\gamma_t \, \mathbb{E}[f(w^t) - f(w^*)] + b\gamma_t^2$$

$$\leq \left(1 - \frac{\mu\gamma_t}{2}\right) \mathbb{E} \left\| w^t - w^* \right\|^2 + b\gamma_t^2. \tag{39}$$

Using $\gamma_t \equiv \gamma$ constant, and since $\gamma \leq \frac{2}{\mu} \Leftrightarrow 1 - \frac{\mu\gamma}{2} \geq 0$ we have by unrolling the recurrence that

$$\mathbb{E} \left\| w^{t+1} - w^* \right\|^2 \leq \left(1 - \frac{\mu\gamma}{2}\right)^{t+1} \left\| w^0 - w^* \right\|^2 + \sum_{k=0}^{t} \left(1 - \frac{\mu\gamma}{2}\right)^k b\gamma^2.$$

Since

$$\sum_{k=0}^{t-1} \left(1 - \frac{\gamma\mu}{2}\right)^t \gamma^2 = 2\gamma^2 \frac{1 - (1 - \gamma\mu/2)^t}{\gamma\mu} \leq \frac{2\gamma^2}{\gamma\mu} = \frac{2\gamma}{\mu},$$

we have that

$$\mathbb{E} \left\| w^{t+1} - w^* \right\|^2 \leq \left(1 - \frac{\mu\gamma}{2}\right)^{t+1} \left\| w^0 - w^* \right\|^2 + \frac{2\gamma b}{\mu},$$

which concludes the proof of (13).

To prove (14) we will apply Theorem 31, for which we will switch our notation to match that of Theorem 31. That is, substituting $\mathcal{L} = \frac{a}{\mu}$, $\sigma^2 = \frac{b}{2}$ and $\hat{\mu} = \frac{\mu}{2}$ into (39) gives

$$\mathbb{E} \left\| w^{t+1} - w^* \right\|^2 \leq (1 - \hat{\mu}\gamma_t) \mathbb{E} \left\| w^t - w^* \right\|^2 + 2\sigma^2\gamma_t^2, \tag{40}$$

which now holds for $\gamma_t \leq \frac{1}{2\mathcal{L}}$ and fits exactly the format of (23), excluding the additional hat on $\mu$. Thus we know from following verbatim the proof that continues after (23) that for a stepsize schedule of

$$\gamma_t = \begin{cases} \dfrac{1}{2\mathcal{L}} & \text{for } t < t^* \\[2ex] \dfrac{1}{\hat{\mu}} \dfrac{2t+1}{(t+1)^2} & \text{for } t \geq t^* \end{cases}$$

for $t^* = 4\lfloor \mathcal{L}/\hat{\mu} \rfloor$ we have that

$$\mathbb{E} \left[\| w^{T+1} - w^* \|^2\right] \leq \frac{16\lfloor \mathcal{L}/\hat{\mu} \rfloor^2}{(T+1)^2} \| w^0 - w^* \|^2 + \frac{\sigma^2}{\hat{\mu}^2} \frac{8}{T+1}.$$

After switching back notation $\mathcal{L} = \frac{a}{\mu}$, $\sigma^2 = \frac{b}{2}$ and $\hat{\mu} = \frac{\mu}{2}$ gives (14). $\qquad\square$

### 9.6 Projected gradient descent with convexity

**Theorem 11.** *Let $\mathcal{W}$ be a nonempty closed convex set. Let $f$ be a convex function, differentiable on $\mathcal{W}$. Let $w^* \in \arg\min_{\mathcal{W}}(f)$. Let $(w^t)_{t\in\mathbb{N}}$ be generated by the Proj-SGD algorithm, with a constant stepsize $\gamma \in (0, +\infty)$. Suppose that $g^t$ is a quadratically bounded estimator (Def. 5) for $\nabla f$ at $w^t$ with constant parameters $(a, b, w^*)$. Then,*

$$\mathbb{E} \left[f(\bar{w}^T) - \inf_{\mathcal{W}} f\right] \leq \frac{\gamma}{2} \left(a \frac{\| w^0 - w^* \|^2}{1 - \theta^T} + b\right).$$

*where $\theta \stackrel{\text{def}}{=} \dfrac{1}{1 + a\gamma^2}$ and $\bar{w}^T \stackrel{\text{def}}{=} \dfrac{\sum_{t=0}^{T-1} \theta^{t+1} w^t}{\sum_{t=0}^{T-1} \theta^{t+1}}$. Finally if $\gamma = \dfrac{\sqrt{2}}{\sqrt{aT}}$ and $T \geq 2$ then*

$$\mathbb{E} \left[f(\bar{w}^T)\right] - \inf f \leq \frac{\sqrt{2a}}{\sqrt{T}} \| w^0 - w^* \|^2 + \frac{b}{\sqrt{2aT}}. \tag{15}$$

*Thus, $T = \mathcal{O}(\epsilon^{-2})$ iterations are sufficient to guarantee that $\mathbb{E} \left[f(\bar{w}^T) - \inf f\right] \leq \epsilon$.*

*Proof.* Plugging in one step of the algorithm, we have that

$$\left\|w^{t+1} - w^*\right\|^2 = \left\|\text{proj}_\mathcal{W}(w^t - \gamma_t g^t) - \text{proj}_\mathcal{W}(w^*)\right\|^2$$
$$\leq \left\|w^t - w^* - \gamma_t g^t\right\|^2,$$

where we used that projections are non-expansive. Taking expectation conditioned on $w^t$, and expanding squares we have that

$$\mathbb{E}\left[\left\|w^{t+1} - w^*\right\|^2 \mid w^t\right] \leq \left\|w^t - w^*\right\|^2 - 2\gamma_t \left\langle \nabla f(w^t),\, w^t - w^*\right\rangle + \gamma_t^2 \mathbb{E}\left[\left\|g^t\right\|^2 \mid w^t\right]$$
$$\leq \left\|w^t - w^*\right\|^2 - 2\gamma_t(f(w^t) - f(w^*)) + a\gamma_t^2 \|w^t - w^*\|^2 + b\gamma_t^2,$$

where we used that $g^t$ is an unbiased estimator with bounded squared norm. Taking full expectation and re-arranging gives

$$2\gamma_t \mathbb{E}\left[f(w^t) - f(w^*)\right] \leq (1 + a\gamma_t^2) \mathbb{E}\left[\left\|w^t - w^*\right\|^2\right] - \mathbb{E}\left[\left\|w^{t+1} - w^*\right\|^2\right] + b\gamma_t^2.$$

From now on we use a constant stepsize $\gamma_t \equiv \gamma$. Let $C = (1 + a\gamma^2)$. Next we would like to setup a telescopic sum. To make this happen, we multiply through by $\alpha_{t+1}$ and impose that $\alpha_{t+1}C = \alpha_t$. This holds with $\alpha_t \overset{\text{def}}{=} \frac{1}{C^t}$. Multiplying through by $\alpha_{t+1}$ and summing from $t = 0, \ldots, T-1$ we have that

$$2\gamma \sum_{t=0}^{T-1} \alpha_{t+1} \mathbb{E}\left[f(w^t) - f(w^*)\right] \leq \sum_{t=0}^{T-1} \left(\alpha_{t+1}C \,\mathbb{E}\left[\left\|w^t - w^*\right\|^2\right] - \alpha_{t+1} \mathbb{E}\left[\left\|w^{t+1} - w^*\right\|^2\right]\right) + \sum_{t=0}^{T-1} \alpha_{t+1}b\gamma^2$$

$$= \sum_{t=0}^{T-1} \left(\alpha_t \,\mathbb{E}\left[\left\|w^t - w^*\right\|^2\right] - \alpha_{t+1} \mathbb{E}\left[\left\|w^{t+1} - w^*\right\|^2\right]\right) + \sum_{t=0}^{T-1} \alpha_{t+1}b\gamma^2$$

$$= \alpha_0 \left\|w^0 - w^*\right\|^2 - \alpha_T \mathbb{E}\left[\left\|w^T - w^*\right\|^2\right] + \sum_{t=0}^{T-1} \alpha_{t+1}b\gamma^2,$$

Using that $\alpha_0 = 1$, dropping the negative term $-\alpha_T \mathbb{E}\left[\left\|w^T - w^*\right\|^2\right]$, dividing through by $2\gamma \sum_{k=0}^{T-1} \alpha_{k+1}$ and using Jensen's inequality we have that

$$\mathbb{E}\left[f(\bar{w}^t) - f(w^*)\right] \leq \frac{\sum_{t=0}^{T-1} \alpha_{t+1} \mathbb{E}\left[f(w^t) - f(w^*)\right]}{\sum_{k=0}^{T-1} \alpha_{k+1}}$$

$$\leq \frac{\left\|w^0 - w^*\right\|^2}{2\gamma \sum_{k=0}^{T-1} \alpha_{k+1}} + \frac{b\gamma}{2}.$$

Finally using that

$$\sum_{k=0}^{T-1} \alpha_{k+1} = \sum_{k=0}^{T-1} C^{-(k+1)} = \frac{1}{C} \sum_{k=0}^{T-1} C^{-k} = \frac{1}{C} \frac{1 - \frac{1}{C^T}}{1 - \frac{1}{C}} = \frac{1 - \frac{1}{C^T}}{C - 1} = \frac{1 - \frac{1}{(1+a\gamma^2)^T}}{a\gamma^2},$$

gives that

$$\mathbb{E}\left[f(\bar{w}^t) - f(w^*)\right] \quad \leq \quad \frac{a\gamma^2}{1 - \frac{1}{(1+a\gamma^2)^T}} \frac{\left\|w^0 - w^*\right\|^2}{2\gamma} + \frac{b\gamma}{2}$$

$$= \quad \frac{a\gamma}{1 - \frac{1}{(1+a\gamma^2)^T}} \frac{\left\|w^0 - w^*\right\|^2}{2} + \frac{b\gamma}{2}$$

$$= \quad \frac{\gamma}{2}\left(\frac{a\left\|w^0 - w^*\right\|^2}{1 - \frac{1}{(1+a\gamma^2)^T}} + b\right)$$

$$= \quad \frac{\gamma}{2}\left(\frac{a\left\|w^0 - w^*\right\|^2}{1 - \theta^T} + b\right)$$

Finally we can apply Lemma 32 with where $B = a$, $A = \sqrt{2/B} = \sqrt{2}/\sqrt{a}$, $\theta = \frac{1}{1+a\gamma^2}$ and $\gamma = A/\sqrt{T} = \sqrt{2}/(\sqrt{a}\sqrt{T})$ which gives

$$
\begin{aligned}
\mathbb{E}\left[f(\bar{w}^T)\right] - \inf f &\leq \frac{1}{2}\frac{A}{\sqrt{T}}\left(2a\|w^0 - w^*\|^2 + b\right) \\
&= \frac{1}{\sqrt{2}}\frac{1}{\sqrt{a}\sqrt{T}}\left(2a\|w^0 - w^*\|^2 + b\right) = \Omega\left(\frac{1}{\sqrt{T}}\right).
\end{aligned}
$$

$\square$

## 10 Solving VI with stochastic optimization

### 10.1 Convergence of the algorithms

**Corollary 12** (Prox-SGD for VI). *Consider the VI problem where $q_w$ is a multivariate Gaussian distribution (Eq. 2) with parameters $w = (m, C) \in \mathbb{R}^d \times \mathcal{T}^d$, and assume that this problem admits a solution $w^*$. Suppose that $\log p(\cdot, x)$ is $M$-smooth and concave (resp. $\mu$-strongly concave). Generate a sequence $w^t$ by using the Prox-SGD algorithm (Def. 6) applied to $l$ and $h$ (Eq. 16), using $\mathsf{g}_{\mathrm{energy}}$ (5) as an estimator of $\nabla l$. Let the stepsizes $\gamma_t$ be constant and equal to $1/(\sqrt{a_{\mathrm{energy}}T})$ (resp. be decaying as in Theorem 7 with $a_{\mathrm{energy}} = 2(d+3)M^2$). Then, for a certain average $\bar{w}^T$ of the iterates, we have for $T \geq 2$ that*

$$
\mathbb{E}\left[f(\bar{w}^T) - \inf f\right] = \mathcal{O}(1/\sqrt{T}) \quad (resp. \; \mathbb{E}\left[\|w^T - w^*\|_2^2\right] = \mathcal{O}(1/T)).
$$

*Proof.* Our assumption on the log target imply that $l$ is smooth and convex (resp. $\mu$-strongly convex) according to Lemmas 18.1 and 19.1. Furthermore, since $\log p(\cdot, x)$ is smooth, we know from Theorem 30 that the gradient estimator $\mathsf{g}_{\mathrm{energy}}$ is quadratically bounded at every $w^t$ with respect to constant parameters $(a, b, w^*)$, with $a = 2(d+3)M^2$. Then, because we work with triangular scale parameters, we know from Lemma 19.3 that $h$ is closed convex. Finally $T \geq \max\{M^2/a, 2\} = 2$ since

$$
\frac{M^2}{a} = \frac{M^2}{2(d+3)M^2} = \frac{1}{2(d+3)} \leq 1.
$$

We have now verified the hypotheses of Theorem 8 (resp. Theorem 7), which gives us the desired bounds. $\square$

**Corollary 13** (Proj-SGD for VI). *Consider the VI problem where $q_w$ is a multivariate Gaussian distribution (Eq. 2) with parameters $w = (m, C) \in \mathbb{R}^d \times \mathcal{S}^d$, and assume that this problem admits a solution $w^*$. Suppose that $\log p(\cdot, x)$ is $M$-smooth and concave (resp. $\mu$-strongly concave). Generate a sequence $w^t$ by using the Proj-SGD algorithm (Def. 9) applied to the function $f = l + h$ (Eq. 16) and the constraint $\mathcal{W}_M$ (Eq. 3), using $\mathsf{g}_{\mathrm{ent}}$ (7) or $\mathsf{g}_{\mathrm{STL}}$ (9) as an estimator of $\nabla f$. Let the stepsizes $\gamma_t$ be constant and equal to $\sqrt{2/(aT)}$ (resp. be decaying as in Theorem 10) with $a = a_{\mathrm{ent}} = 4(d+3)M^2$ or $a = a_{\mathrm{STL}} = 24(d+3)M^2$. Then, for a certain average $\bar{w}^T$ of the iterates, we have for $T \geq 2$ that*

$$
\mathbb{E}\left[f(\bar{w}^T) - \inf f\right] = \mathcal{O}(1/\sqrt{T}) \quad (resp. \; \mathbb{E}\left[\|w^T - w^*\|_2^2\right] = \mathcal{O}(1/T)).
$$

*Proof.* Our assumption on the log target imply that $l$ is smooth and convex (resp. $\mu$-strongly convex) according to Lemmas 18.1 and 19.1. Regarding the entropy, since $\log p(\cdot, x)$ is $M$-smooth, we know from Lemma 18.3 that $h$ is $M$-smooth on $\mathcal{W}_M$. We also know that $h$ is closed convex, since we consider symmetric scale parameters (see Lemma 19.2). Furthermore, the smoothness of $\log p(\cdot, x)$ implies that the gradient estimator $\mathsf{g}_{\mathrm{ent}}$ is quadratically bounded at every $w^t \in \mathcal{W}_M$, with respect to constant parameters $(a, b, w^*)$. This is stated in Theorem 30, where we can see that for $\mathsf{g}_{\mathrm{ent}}$ we can take $a_{\mathrm{ent}} = 4(d+3)M^2$, and for $\mathsf{g}_{\mathrm{STL}}$ we can take $a_{\mathrm{STL}} = 2(d+3)(2K^2 + (2M)^2)$. Since we know from Theorem 4 that $K \leq 2M$, we can consider that $a_{\mathrm{STL}} \leq 24(d+3)M^2$. Finally, we know from Lemma 20 that the minimizers of $f$ belong to $\mathcal{W}_M$, meaning that $\mathrm{argmin}(f) = \mathrm{argmin}_{\mathcal{W}_M}(f)$. We have now verified the hypotheses of Theorem 11 (resp. Theorem 10), which gives us the desired bounds. $\square$

**Corollary 14** (Proj-SGD for VI - Gaussian target). *Consider the setting of Corollary 13, in the scenario that $\log p(\cdot, x)$ is $\mu$–strongly concave, that we use the $g_{\text{STL}}$ estimator, and that we take a constant stepsize $\gamma_t \equiv \gamma \in \left(0, \min\{\frac{\mu}{2a_{\text{STL}}}, \frac{2}{\mu}\}\right]$. Assume further that $p(\cdot|x)$ is Gaussian. Then,*

$$\mathbb{E}\left[\|w^T - w^*\|_2^2\right] \leq \left(1 - \frac{\mu\gamma}{2}\right)^T \|w^0 - w^*\|_2^2.$$

*Proof.* We use the same arguments as in the proof of Corollary 13, but this time we use a constant stepsize, which gives us the bound (13) from Theorem 10, where $b = 4(d+3)K^2\|w^* - \hat{w}\|^2$ according to Theorem 4. All we need to conclude is to verify that $b = 0$. This comes from the assumption that the target is Gaussian, from which we deduce that $K = 0$ (see again Theorem 4). $\qquad\square$

## 10.2 Computations for the algorithms

**Lemma 35** (Prox of the entropy for triangular matrices). *Assume that $V = \mathcal{T}^d$, and let $w = (m, C) \in \mathbb{R}^d \times \mathcal{T}^d$. Then $\text{prox}_{\gamma h}(w) = (m, \hat{C})$, where*

$$\hat{C}_{ij} = \begin{cases} \frac{1}{2}(C_{ii} + \sqrt{C_{ii}^2 + 4\gamma}) & \text{if } i = j, \\ C_{ij} & \text{if } i \neq j. \end{cases}$$

*Proof.* This is essentially proved in [13, Theorem 13]. We provide a proof here for the sake of completeness, also because our entropy here enforces positive definiteness. So we need to compute

$$\operatorname*{argmin}_{z \in \mathbb{R}^d, X \in \mathcal{T}^d, X \succ 0} - \log \det X + \frac{1}{2\gamma}\|z - m\|^2 + \frac{1}{2\gamma}\|X - C\|_F^2.$$

Because the objective here is separated in $z$ and $X$, we can optimizae those variables separately. It is clear that the optimal $z$ will be $z = m$. For the matrix component, it remains to solve

$$\operatorname*{argmin}_{X \in \mathcal{T}^d, X \succ 0} - \log \det X + \frac{1}{2\gamma}\|X - C\|_F^2.$$

Now remember that for a triangular matrix, being positive definite is equivalent to have positive diagonal elements (Lemma 19.3). Because $X$ is a lower triangular matrix, we have $-\log \det X = \sum_{i=1}^d -\log X_{ii}$ and $X_{ij} = 0$ when $i < j$. So our problem becomes

$$\operatorname*{argmin}_{X \in \mathcal{T}^d, X_{ii} > 0} \sum_{i=1}^d (-\log X_{ii}) + \sum_{i \geq j} \frac{1}{2\gamma}(X_{ij} - C_{ij})^2.$$

This is a gain a separated optimization problem with respect to the variables $X_{ij}$. So we deduce that when $i \geq j$ we must take $\hat{C}_{ij} = C_{ij}$, and when $i = j$ we need to take

$$\hat{C}_{ii} = \operatorname*{argmin}_{x > 0} - \log x + \frac{1}{2\gamma}(x - C_{ii})^2. \tag{41}$$

This is the proximal operator of $-\log$, which can be classically computed [55, Example 6.9], and gives the desired result. $\qquad\square$

**Lemma 36** (Projection onto $\mathcal{W}_M$ for symmetric matrices). *Assume that $V = \mathcal{S}^d$, and let $w = (m, C) \in \mathbb{R}^d \times \mathcal{S}^d$. Let $C = UDU^\top$ be the eigenvalue decomposition of $C$, with $D$ diagonal and $U$ orthogonal. Then $\text{proj}_{\mathcal{W}_M}(w) = (m, U\tilde{D}U^\top)$, where $\tilde{D}$ is the diagonal matrix defined by*

$$\tilde{D}_{ii} = \max\{D_{ii}, \frac{1}{\sqrt{M}}\}.$$

*Proof.* Consider $g = \delta_{\mathcal{W}_M}$ to be the indicator function of $\mathcal{W}_M$, which is equal to 0 on $\mathcal{W}_M$ and $+\infty$ elsewhere. Then $\text{proj}_{\mathcal{W}_M} = \text{prox}_g$, and we can compute it with [55, Theorem 7.18]. For this, we need to write $g(C) = F(\lambda(C))$, where $\lambda(C)$ is the set of eigenvalues of $C$, and $F$ is the indicator function of $K := \{z \in \mathbb{R}^d : z_i \geq 1/\sqrt{M}\}$. In that way, we see that $g$ is a symmetric spectral function in the sense of [55, Definition 7.12], and the conclusion follows. $\qquad\square$

**Lemma 37** (Adjoint of the distribution transformation). *Let $u \in \mathbb{R}^d$ be fixed. Let $T : \mathbb{R}^d \times V \to \mathbb{R}^d$ be such that $T(m, C) = Cu + m$. Then $T$ is a linear operator such that its adjoint $T^* : \mathbb{R}^d \to \mathbb{R}^d \times V$ verifies $T^*(z) = (z, \text{proj}_V(zu^\top))$.*

*Proof.* The fact that $T$ is linear is immediate. Let us consider now the extension $\bar{T} : \mathbb{R}^d \times \mathcal{M}^d \to \mathbb{R}^d$ such that $\bar{T}(m, C) = Cu + m$. If we define the canonical injection $\iota : \mathbb{R}^d \times V \to \mathbb{R}^d \times \mathcal{M}^d$, we clearly have $T = \bar{T} \circ \iota$. This means that $T^* = \iota^* \circ \bar{T}^* = \text{proj}_{\mathbb{R}^d \times V} \circ \bar{T}^*$. It remains to prove that $\bar{T}^*(z) = (z, zu^\top)$. For this, it suffices to use the properties of the trace to write that

$$
\begin{aligned}
\langle (z, zu^\top), (m, C) \rangle &= \langle z, m \rangle + \langle zu^\top, C \rangle = \langle z, m \rangle + \text{tr}(uz^\top C) \\
&= \langle z, m \rangle + \text{tr}(z^\top Cu) = \langle z, m \rangle + \langle z, Cu \rangle \\
&= \langle z, T(m, C) \rangle.
\end{aligned}
$$

$\square$

### 10.3 Explicit implementation of the algorithms

**Lemma 38** (Explicit computation of the estimators). *Let $u \in \mathbb{R}^d$, and let $w = (m, C) \in \mathbb{R}^d \times V$ where $C \succ 0$. Define*

$$
\pi := -\nabla_z \log p(Cu + m, x).
$$

*Then*

1. $g_{\text{energy}}(u) = (\pi, \text{proj}_V(\pi u^\top))$,

2. $g_{\text{ent}}(u) = (\pi, \text{proj}_V(\pi u^\top - C^{-\top}))$,

3. $g_{\text{STL}}(u) = (\pi - C^{-\top} u, \text{proj}_V(\pi u^\top - C^{-\top} uu^\top))$.

*Proof.* Let $w = (m, C)$ and let $T$ be the linear operator defined by $T(m, C) = Cu + m$, as in Lemma 37. As a shorthand we will note $p_x := p(\cdot, x)$.

1. From its definition in (5), the energy estimator is equal to

$$
g_{\text{energy}}(u) = -\nabla_w (\log \circ p_x \circ T)(w).
$$

Applying the chain rules, we obtain

$$
g_{\text{energy}}(u) = -T^* \nabla_z (\log \circ p_x)(Tw) = T^* \pi.
$$

The conclusion follows from the expression of $T^*$ obtained in Lemma 37.

2. From its definition in (7), the entropy estimator $g_{\text{ent}}(u)$ is equal to $g_{\text{energy}}(u) + \nabla h(w)$, where (we use the fact that we assumed that $w \in \mathcal{W}$) we know from Lemma 18.2 that $\nabla h(w) = -(0, \text{proj}_V(C^{-\top}))$. The conclusion follows from the previous item and the linearity of $\text{proj}_V$.

3. From its definition in (9), the STL estimator is equal to

$$
g_{\text{STL}}(u) = g_{\text{energy}}(u) + \nabla_w(\phi \circ T)(w),
$$

where $\phi(z) = \log q_v(z)$, with $v$ being a copy of $w$. From the definition of a Gaussian density (17) we have that

$$
\nabla_z \phi(z) = -(C^{-\top} C^{-1})(z - m),
$$

so we deduce that

$$
\nabla_w(\phi \circ T)(w) = T^* \nabla_z \phi(Tw) = -T^*(C^{-\top} C^{-1})(Tw - m) = -T^* C^{-\top} u.
$$

The conclusion follows after combining the first item with Lemma 37.

$\square$

**Lemma 39** (Pseudocode of the algorithms).

1. *Algorithm 1 is equivalent to the Prox-SGD algorithm (Def. 6) applied to $l$ and $h$ (Eq. 16), using $\mathbf{g}_{\text{energy}}$ (5) as an estimator of $\nabla l$ and using lower triangular covariance factors.*

2. *Algorithm 2 is equivalent to the Proj-SGD algorithm (Def. 9) applied to the function $f = l + h$ (Eq. 16) and the constraint $\mathcal{W}_M$ (Eq. 3), and using $\mathbf{g}_{\text{ent}}$ (7) as an estimator of $\nabla f$.*

3. *Algorithm 3 is equivalent to the Proj-SGD algorithm (Def. 9) applied to the function $f = l + h$ (Eq. 16) and the constraint $\mathcal{W}_M$ (Eq. 3), and using $\mathbf{g}_{\text{STL}}$ (9) as an estimator of $\nabla f$.*

*Proof.* In this proof, we fix an iteration $t \in \mathbb{N}$, consider $u_t \sim \mathcal{N}(0, I)$ and we note $w^t = (m_t, C_t)$.

1. From (Def. 6) we know that $w^{t+1} = \text{prox}_{\gamma_t h}(w^t - \gamma_t \mathbf{g}_{\text{energy}}(u_t))$. According to Lemma 38, we have
$$w^t - \gamma_t \mathbf{g}_{\text{energy}}(u_t) = (m_t, C_t) - \gamma_t(\pi_t, \text{proj}_{\mathcal{T}^d}(\pi_t u_t^\top)),$$
where
$$\pi_t = -\nabla_z \log p(C_t u_t + m_t, x).$$
Moreover, we know from Lemma 35 what $\text{prox}_{\gamma_t h}$ is:
$$\text{prox}_{\gamma_t h}(m, C) = (m, \mathcal{R}_{\gamma_t}(C)), \text{ where } \mathcal{R}_{\gamma_t}(C)_{ij} = \begin{cases} \frac{1}{2}(C_{ii} + \sqrt{C_{ii}^2 + 4\gamma_t}) & \text{if } i = j, \\ C_{ij} & \text{if } i \neq j. \end{cases}$$
First, we see that with respect to the $m$ variable the proximal operator is the identity, meaning that we indeed have
$$m_{t+1} = m_t - \gamma_t \pi_t.$$
Second, we see that with respect to the $C$ variable we have
$$C_{t+1} = \mathcal{R}_{\gamma_t}(C_t - \gamma_t \text{proj}_{\mathcal{T}^d}(\pi_t u_t^\top)).$$
It is immediate to see by induction that since we initialize with $C_0 \in \mathcal{T}^d$, then $C_t \in \mathcal{T}^d$ as well. Therefore we can use the linearity of $\text{proj}_{\mathcal{T}^d}$ to write that
$$C_{t+1} = \mathcal{R}_{\gamma_t}\left(\text{proj}_{\mathcal{T}^d}(C_t - \gamma_t \pi_t u_t^\top)\right).$$
The conclusion follows after observing that
$$(\mathcal{R}_{\gamma_t} \circ \text{proj}_{\mathcal{T}^d})(C)(i, j) = \begin{cases} C(i, j) & \text{if } i > j, \\ \frac{1}{2}\left(C(i, i) + \sqrt{C(i, i)^2 + 4\gamma_t}\right) & \text{if } i = j, \\ 0 & \text{if } i < j. \end{cases}$$

2. From (Def. 9) we know that $w^{t+1} = \text{proj}_{\mathcal{W}_M}(w^t - \gamma_t \mathbf{g}_{\text{ent}}(u_t))$. According to Lemma 38, we have
$$w^t - \gamma_t \mathbf{g}_{\text{ent}}(u_t) = (m_t, C_t) - \gamma_t(\pi_t, \text{proj}_{\mathcal{S}^d}(\pi_t u_t^\top - C_t^{-\top})).$$
Moreover, we know from Lemma 36 what $\text{proj}_{\mathcal{W}_M}$ is:
$$\text{proj}_{\mathcal{W}_M}(m, C) = (m, \tilde{\mathcal{R}}(C)),$$
where $\tilde{\mathcal{R}}(C) = U\tilde{D}U^\top$, with $UDU^\top$ being a SVD decomposition of $C$, and $\tilde{D}(i, i) = \max\{D(i, i); M^{-1/2}\}$. First, we see that with respect to the $m$ variable the projection is the identity, meaning that we indeed have
$$m_{t+1} = m_t - \gamma_t \pi_t.$$
Second, we see that with respect to the $C$ variable we have
$$C_{t+1} = \tilde{\mathcal{R}}(C_t - \gamma_t \text{proj}_{\mathcal{T}^d}(\pi_t u_t^\top - C_t^{-\top})).$$
It is immediate to see by induction that since we initialize with $C_0 \in \mathcal{S}^d$, then $C_t \in \mathcal{S}^d$ as well. Therefore we can use the linearity of $\text{proj}_{\mathcal{S}^d}$ to write that
$$C_{t+1} = \tilde{\mathcal{R}}\left(\text{proj}_{\mathcal{S}^d}(C_t - \gamma_t(\pi_t u_t^\top - C_t^{-\top}))\right).$$
The conclusion follows after setting $\hat{C}_{t+1} = C_t - \gamma_t(\pi_t u_t^\top - C_t^{-\top})$, and using the fact that $\text{proj}_{\mathcal{S}^d}(C) = (C + C^\top)/2$.

3. Use the same reasoning as in the previous item, with the appropriate computation for the STL estimator from Lemma 38.

$\square$

---

**Algorithm 1** Prox-SGD with energy estimator and triangular factors

---

Let $m_0 \in \mathbb{R}^d$, $C_0 \in \mathcal{T}^d$ such that $C_0 \succ 0$, $(\gamma_t)_{t \in \mathbb{N}} \subset (0, +\infty)$
**for** $t \in \mathbb{N}$ **do**
    Sample $u_t \sim \mathcal{N}(0, I)$
    Compute $\pi_t = -\nabla_z \log p(C_t u_t + m_t, x)$
    Set $m_{t+1} = m_t - \gamma_t \pi_t$
    Set $\hat{C}_{t+1} = C_t - \gamma_t \pi_t u_t^\top$
    Set $C_{t+1}$ via $C_{t+1}(i,j) = \begin{cases} \hat{C}_{t+1}(i,j) & \text{if } i > j \\ \frac{1}{2}\left( \hat{C}_{t+1}(i,i) + \sqrt{\hat{C}_{t+1}(i,i)^2 + 4\gamma_t} \right) & \text{if } i = j \\ 0 & \text{if } i < j \end{cases}$

---

---

**Algorithm 2** Proj-SGD with entropy estimator and symmetric factors

---

Let $m_0 \in \mathbb{R}^d$, $C_0 \in \mathcal{S}^d$ such that $C_0 \succ 0$, $(\gamma_t)_{t \in \mathbb{N}} \subset (0, +\infty)$
**for** $t \in \mathbb{N}$ **do**
    Sample $u_t \sim \mathcal{N}(0, I)$
    Compute $\pi_t = -\nabla_z \log p(C_t u_t + m_t, x)$
    Set $m_{t+1} = m_t - \gamma_t \pi_t$
    Set $\hat{C}_{t+1} = C_t - \gamma_t (\pi_t u_t^\top - C_t^{-1})$
    Compute $U_{t+1} \hat{D}_{t+1} U_{t+1}^\top$ a singular value decomposition of $(\hat{C}_{t+1} + \hat{C}_{t+1}^\top)/2$
    Set $C_{t+1} = U_{t+1} D_{t+1} U_{t+1}^\top$, where $D_{t+1}(i,i) = \max\{\hat{D}_{t+1}(i,i); M^{-1/2}\}$

---

---

**Algorithm 3** Proj-SGD with STL estimator and symmetric factors

---

Let $m_0 \in \mathbb{R}^d$, $C_0 \in \mathcal{S}^d$ such that $C_0 \succ 0$, $(\gamma_t)_{t \in \mathbb{N}} \subset (0, +\infty)$
**for** $t \in \mathbb{N}$ **do**
    Sample $u_t \sim \mathcal{N}(0, I)$
    Compute $\pi_t = -\nabla_z \log p(C_t u_t + m_t, x)$
    Set $m_{t+1} = m_t - \gamma_t (\pi_t - C_t^{-1} u_t)$
    Set $\hat{C}_{t+1} = C_t - \gamma_t (\pi_t u_t^\top - C_t^{-1} u_t u_t^\top)$
    Compute $U_{t+1} \hat{D}_{t+1} U_{t+1}^\top$ a singular value decomposition of $(\hat{C}_{t+1} + \hat{C}_{t+1}^\top)/2$
    Set $C_{t+1} = U_{t+1} D_{t+1} U_{t+1}^\top$, where $D_{t+1}(i,i) = \max\{\hat{D}_{t+1}(i,i); M^{-1/2}\}$

---

