# OpenReview forum: "Provable convergence guarantees for black-box variational inference"
_NeurIPS.cc/2023/Conference — NeurIPS 2023 poster_

### Official Review · Reviewer_vBBM · 2023-07-03

**Soundness:** 3 good
**Presentation:** 3 good
**Contribution:** 4 excellent
**Rating:** 7
**Confidence:** 3

**Summary:**

This paper offers the first convergence results for black-box VI, a widely used and popular framework for Bayesian problems. Under assumptions on the log model, $\log p$, and given a Gaussian variational family of distributions, convergence rates are established by utilizing recent advances in the field. The assumptions are motivated by giving practical examples of when they are satisfied.

**Strengths:**

This work seems to lay the missing puzzle piece in a series of works towards establishing convergence rates for black-box VI. As such, it is clearly original, and it is significant as black-box VI is widely used in practice and is a popular research topic. Although the results build on seemingly strong assumptions on the log model, $\log p$, the significance of the results is emphasized by exemplifying broad classes of problem settings where the assumptions are satisfied, which deserves credit. I believe this paper will be of interest to the NeurIPS community.

The paper follows a clear line of argument and builds up to its results in a pedagogical manner, giving a convincing account of the existing, related works.

**Weaknesses:**

**W1**: $C$ is a matrix (line 63), but in line 120 I read it as $C = 0$, by inspecting $\bar{w} = (\bar{m}, 0)$. It is not clear to me what this means. Are all elements in the covariance matrix zero (since $\Sigma = C C^T$)?

**W2**: In Sec. 2.1 I read it as, if $\log p$ is $M$-smooth then we can expect that the optimal covariance parameter for $q_w$ should not be too small. This makes sense to me. In fact they should be at least $1 / \sqrt{M}$ as shown in Domke's previous work. However in Theorem 2, where the log-target is assumed to be $M$-smooth, it says that $\bar{w}=(\bar{m}, 0)$ is the "maximum of $\log p$". I am confused about this. Doesn't $\bar{w}$ here imply that $\log p$ is not smooth, and that the optimal parameters $w^* = \bar{w}$, i.e. $C^* = 0 < 1 / \sqrt{M}$?

If I am helped clear up this confusion, I am willing to raise my score.

**W3:** I am not sure how the discussion about the score-type estimator in Eq. 4 contributes to the paper. The results in paper are based on the reparameterization trick (i.e. the path-type estimator in Eq. 5), no? What is the contribution of this discussion to the results in the paper?

**Minor issues**

In line 16, $h$ is called the entropy, but as it is defined in Eq. 1 and 15 it is the neg-entropy.

Sometimes (e.g. in the Central contributions box and on line 93) $q_w$ is referred to a family of distributions and sometimes (line 243) it is used a density. In VI the "family" is often meant as the set of distributions in which we want to find the optimal approximation, no? So can it really be a family and a distribution simultaneously? Perhaps it is more clear if $\mathcal{Q}_w$ is used to notate the family of Gaussian distributions parameterized by $w$?

Typo:
* line 35: objective is misspelled
* line 141: "that bar" seems to be a typo

**Questions:**

* In Sec. 2.1 it stated that $h$ is $M$-smooth over the set $\mathcal{W}_M$. Based on this, do you make any assumptions on $q_w$ other than it being Gaussian? I was expecting some new assumption given the title of the section, but maybe this is a "structural property"?
* Out of curiosity, could you expand on the constructed problem where the variance of the path-based estimator is "vastly" increased and the score-based estimator is not? Where is the noise added? To $\phi$?

**Limitations:**

The assumptions are clearly highlighted and discussed in the paper.

---

> ### Author Rebuttal · Authors · 2023-08-08
>
> ### C is a matrix but C=0
>
> Yes, in this context, of writing something like w=(m,0) then 0 means a matrix of zeros. If φ(z)=-log p(z,x), then the gradient noise is bounded in terms of how far the parameters w=(m,C) are from representing a delta function centered at the MAP solution. (Note this is different from the optimal solution w*.)
>
> ### $\bar{w}$ is the maximum of $\log p$, doesn't this imply log p is not smooth?
>
> It’s important to note here that $\bar{w}$ is not the optimal solution to the problem, which we denote by $w^*$. We do not say that $\bar{w}$ is the maximum of $\log p$ (which is not true) but rather that $\bar{w} = (\bar m, 0)$, where $\bar m$ is the maximum of $\log p$. Intuitively, we can see $\bar{w}$ as parameters that represent a delta function (because it has zero covariance) centered at the MAP solution. Note that this is just an intuition and we never make use of this interpretation. We propose to rewrite the Theorem to make this more clear.
>
> ### Discussion of score-type estimators
>
> We appreciate this point. Arguably this discussion is unnecessary and distracting at the point it currently happens. Our reason for including it was that we wanted to mention that there are situations where score-function type estimators have lower variance than reparameterization, and so these arguably deserve further investigation. We believe this comment would make more sense in the discussion section.
>
> ### Any other assumptions needed for h to be M-smooth?
>
> The assumption that h is M-smooth (over $W_M$) requires no other assumptions : it is indeed a structural property. We will clarify how this is discussed.
>
> ### Expand on the problem with variance of path-based is increased and score-based is not?
>
> In terms of how the variance of the path-based estimator could be increased, imagine some problem with an unnormalized posterior φ(z) which is equal (up to a constant) to -log p(z,x). Suppose for simplicity that z is just a scalar. Now imagine changing to a new unnormalized posterior φ’(z) = φ(z) + √ε sin(z/ε) for some very small value ε. The difference of φ and φ’ is trivial (they only vary by a tiny amount). However, the derivative of the added term is cos(z/ε) / √ε. When ε is very small, this derivative will be huge. This will vastly increase the noise of a reparameterization estimator but have little effect on a score-based estimator. (In a sense, this is why the assumption that log p(z,x) is M-smooth is so important—it prohibits posteriors that do things like this!)
>
> ### errors / typos
>
> Thank you for pointing these out. We will correct them. We agree that the way we used the terminology of “density” vs “family” is confusing. We will fix this by saying that q_w for a specific w is a Gaussian distribution, whereas the set {q_w | w ∈ W} (or Q_w)  is the family.

---

> > ### Comment · Reviewer_vBBM · 2023-08-15
> > **Response to rebuttal**
> >
> > Thank you for the clarifications provided in the rebuttal.
> >
> > Especially I appreciated the clarifications and the example regarding the score-type estimator and the variance of the path-based estimator. Now that I can more clearly see the point of the discussion, I think it makes a nice point why the assumption of an M-smooth log-density is important. I would be glad if it was somehow included in the discussion section.
> >
> > I will increase my score to 7 if the issue in the proof raised by gXey is fixed.

---

### Official Review · Reviewer_gXey · 2023-07-04

**Soundness:** 4 excellent
**Presentation:** 4 excellent
**Contribution:** 3 good
**Rating:** 8
**Confidence:** 4

**Summary:**

This paper offers a convergence proof for the stochastic optimization problem inherent in full-rank Gaussian variational inference when the log-density of the target is concave. The primary challenge of the convergence proof lies in managing the non-smoothness present in the entropy term of Gaussian VI. This issue is addressed in the current work by considering proximal and projected stochastic gradient descent. To evaluate the validity of the imposed assumptions, case studies involving Bayesian (generalized) linear regression problems are provided.

**Strengths:**

I appreciate the clarity of the writing of the present work; it clearly descibes the scope of the problem studied and states the main challenge of showing the convergence. The layout of each section forms a smooth flow of the proof strategies, in addition to a nice organization of the mathematical proofs, which makes the reading enjoyable.

The case study provided in appendix (sec 7.3) is helpful for reasoning imposed assaumptions.

The proximal operator introduced for optimizing the covariance matrix is novel to me. Although it's a standard techniques in optimization literature---splitting non-strongly smooth (but closed convex) part of the objective function using proximal operator, it's the first time I see in Gaussian VI literature. (This could of course be due to my lack of knowledge in this field) Additionally, the weighted telescope summation used when $-logp$ is only convex(rather than strongly convex) is very interesting, which is a setting that rarely considered in literature as far as I'm aware.

Even though the paper limits the scope to dense Gaussian VI problem, I think the present techniques can be applied to general location-scale variational families (as long as the variance quadratic bounds still holds).



**Weaknesses:**

In my view, the principal limitation of this work is the somewhat restricted scope of the Variational Inference (VI) problem it examines. Specifically, the variational family under consideration is: (1.) Gaussian (albeit full-rank), and (2.) the target distribution is log-concave or strongly-log-concave. Additionally, (3.) data subsampling on $\log p$ is not taken into account. The proof techniques necessitated by this setting, in my opinion, are fairly standard within the stochastic optimization literature. While I acknowledge that any relaxation of condition (2) would likely preclude anything beyond convergence to some stationary point, and that the quadratic variance bound is heavily reliant on condition (1) (which is probably not a major concern for the broader community in regards to more complex VI families), offering a convergence result when data subsampling is implemented could greatly increase the impact of this work. I would be inclined to raise the score to 8 if data subsampling was considered, or if existing proof techniques could easily address it (though I doubt the current approach is effective in this case, please correct me if I'm mistaken).

Regarding originality, even though I concur that explicit results may be lacking for this specific problem setting, I don't think this is the **first** optimization theory guarantee for full-rank Gaussian VI. For instance, [Xu & Campbell, 2022] studies the convergence of full-rank Gaussian VI without the assumption of a log-concave target (even though they utilize posterior asymptotics to somewhat reduce the underlying optimization problem on a strongly-log-concave target, they also employ a special scaling operator (Eq(8)) to handle the non-Lip-smoothness of the entropy term $\log \text{det}$). I recommend that the authors integrate an optimization analysis of the scaled stochastic gradient descent proposed in [Xu & Campbell, 2022], and carefully compare the results to those provided in [Xu & Campbell, 2022] within a strongly convex setting at least (ignoring the data asymptotics and merely assuming that $\log p$ is strongly-log-concave should align their work with the current setting).

Finally,  I encourage the authors to provide additional case studies, perhaps involving more Bayesian GLMs or even Bayesian sparse regression with a Horseshoe prior. From what I understand, most of these models yield a log-concave target, but the application of the variance quadratic bound is unclear to me. If negative results were to appear, for example, some models not having a quadratically bounded gradient noise, these examples would still significantly benefit the community.


At the end, I hope to stress that despite the aforementioned weakness/limitations, I think this is a really good work for this niche research direction.

Reference:
 [Xu&Campbell, 2022]: The computational asymptotics of Gaussian variational inference and the Laplace approximation, Statistics and Computing

**Questions:**

1. Compare to  [Xu&Campbell, 2022]: As I mentioned earlier, this is a very relavant past work on this topic. I wonder specifically, if we ignore all the asymptotic stuff and assumes $\log p$ is strongly-log-concave, does [Xu&Campbell, 2022] achieve similar convergence rate ($O(1/T)$) as the present paper?

2. I wonder if the proximal operator (line 264) has been noted in past VI literature? Or it is proposed by the author.



**Limitations:**

See Weakness

---

> ### Author Rebuttal · Authors · 2023-08-08
>
> Thank you very much for your review.
>
> ### Convergence result when data subsampling
>
> Our proofs can indeed address subsampling with minor technical difficulty, namely bounding the variance of a slightly changed gradient estimator. Note that the main optimization results only depend on (1) the structural properties of the problem (which are unchanged with subsampling) and (2) the constants (a,b) bounding the gradient variance. Our gradient variance guarantees in Theorems 2 and 3 are all based on Theorem 1 from [20]. However, that reference also provides a more general version of Theorem 1 (Theorem 6 in [20]) which considers data subsampling. Very roughly speaking, with uniform data subsampling of 1 datum at a time, the expected squared norm of the gradient increases by a factor of between $1$ (no increase) and $ndata$, depending on how correlated the data are. (The less correlation, the larger the increase.) So, with subsampling, both Theorem 2 and Theorem 3 would increase by a factor of $1$ to $ndata$. (To be more precise, the second term in theorem 3 would be unchanged, though this would typically be dominated by the first term.) Those increases would then manifest as an increase of between $1$ and $ndata$ for both a and b in the quadratic noise bounds. After that point, exactly the same convergence results hold. We will add a discussion of this.
>
> ### Compare to [Xu&Campbell]
>
> Thank you for pointing out this paper which we were not aware of. It does indeed provide a convergence guarantee for an algorithm similar to proximal-SGD using the g_{energy} estimator, and it was an oversight not to cite and discuss it. The principal differences with our work is (1) their guarantee is asymptotic (holds in the limit of large T) (2) their guarantee is local (holds when started close to the solution) (3) our analysis also considers the g_{ent} estimator, and (4) our analysis considers the convex and strongly-convex cases, whereas theirs considers the non-convex case . Their analysis does not consider the strongly convex (equivalently when p is strongly-log-concave) assumption so only gives a O(1/√T) rate [Thm 3 in their paper] rather than O(1/T). We will discuss this paper and revise our claims appropriately.
>
> ### Prox operator
>
> The proximal operator has indeed been used in the past in VI [13] which we should note more clearly.

---

> > ### Comment · Reviewer_gXey · 2023-08-14
> >
> > Thank you for your reply. I believe the authors response has addressed all of my questions properly, so I'd love to raise the score to 8.
> > However, I found a mistake in the proof of Lemma 28, which could potentially cause some issue.
> >
> > Line 670 reads:
> > $$\frac{1}{1-\theta^T} \leq \frac{1}{(1-\theta) T} = ...$$ Here $\theta:=\frac{1}{1+B \gamma^2} \in (0, 1)$.
> > Note that the LHS of the above inequality converges to $1$ as $T\to \infty$, while the RHS converges to $0$, which is incorrect. Although I don't think the proof is not fixable -- intuitively Lemma 8 gives the sensible rate, it's important to have the proof corrected since Lemma 28 is a key step for convergence results in convex cases (Thm7 and Thm 10), which I consider to be the key contribution of the paper.
> >
> > Therefore, I'll keep my current score. If the proof is fixed, I'll update the score to 8, otherwise I'll reduce the score to 6.

---

> > > ### Author Response · Authors · 2023-08-16
> > > **Fixed issue**
> > >
> > > Dear Reviewer,
> > >
> > > thank you very much for spotting this mistake in the proof. Fortunately, it is easy to fix (we provide details for you below) and this fix has no consequences on the results, besides a few multiplicative constants being modified.
> > > The key is to prove that $1/(1-\theta^T)$ will be less than 2, provided $T \geq 2$ and that the stepsize is taken as $\gamma= A/\sqrt{T}$, for an appropriate choice of A. This is encapsulated in the revised Lemma which is below in a separate comment below.

---

> > > > ### Author Response · Authors · 2023-08-16
> > > > **Lemma 28 (revised)**
> > > >
> > > > Suppose we are given a sequence $\bar{w}^T$ and constants $a,b,B>0$ such that
> > > > $$
> > > > E[f(\bar{w}^T)]-\inf f  \leq  \gamma \left(\frac{a\|w^0-w^*\|^{2}}{1-\theta^{T}}+b\right)
> > > > $$
> > > > holds for $$ \theta  := \\frac{1}{1+B\gamma^{2}}. $$
> > > > If $\gamma=\frac{A}{\sqrt{T}}$ for $A \geq \sqrt{\frac{2}{B}}$ and $T \geq 2$, then
> > > > $$  E[f(\bar{w}^T)]-\inf f  \leq \frac{A}{\sqrt{T}}(2a\| w^0-w^*\|^2 +b).$$
> > > > Thus, $T=\mathcal{O}(\epsilon^{-2})$ iterations are sufficient
> > > > to guarantee that $E[f(\bar{w}^{T})]-\inf f\leq\epsilon.$
> > > >
> > > > The proof is in the next comment
> > > >
> > > > **Note:** We note that once applied to the VI problem, this means that the stepsize $\gamma$ must be taken equal to
> > > > $\gamma= 1/\sqrt{aT}$ for ProxSGD, and $\gamma= \sqrt{2/aT}$ for ProjSGD.

---

> > > > > ### Author Response · Authors · 2023-08-16
> > > > > **Proof of Lemma 28**
> > > > >
> > > > > Proof:
> > > > >
> > > > >
> > > > > First we show that $\frac{1}{1-\theta^T} \leq 2$ is  equivalent to
> > > > > \begin{equation}
> > > > > T \geq \frac{\log 2}{\log(1+BA^2/T)}.
> > > > > \end{equation}
> > > > > Indeed this follows since
> > > > > \begin{align*}
> > > > >     \frac{1}{1-\theta^T} &\leq 2 \quad  \Leftrightarrow \\\\
> > > > >     \theta^T & \leq \frac{1}{2}   \quad \Leftrightarrow \\\\
> > > > >     % T \log \theta & \leq \log \frac{1}{2}  \quad \Leftrightarrow \\\\
> > > > >      T  & \geq \frac{\log 2}{\log 1/\theta} \quad \Leftrightarrow \\\\
> > > > >   T  & \geq \frac{\log 2}{\log (1+B\gamma^2)} \quad \Leftrightarrow \\\\
> > > > >  T  & \geq \frac{\log 2}{\log (1+BA^2/T)}.
> > > > >   \end{align*}
> > > > > Now note that $\log(1+x)\geq \frac{x}{1+x}$ for $x \geq 0$, which implies that $\frac{1}{\log(1+x)}\leq 1+\frac{1}{x}$ for $x \geq 0$. Applying this   gives that
> > > > > \begin{eqnarray*}
> > > > > \frac{1}{\log(1+BA^2/T)} \leq 1 + \frac{T}{BA^2}.
> > > > > \end{eqnarray*}
> > > > > So to guarantee $\frac{1}{1-\theta^T} \leq 2$, from the above it is sufficient to enforce that
> > > > > \begin{eqnarray*}
> > > > > T & \geq &  1 + \frac{T}{BA^2}\\
> > > > > &\geq & \log (2) \left( 1 + \frac{T}{BA^2} \right). \\
> > > > > \end{eqnarray*}
> > > > > Since we assumed that $A \geq \sqrt{2/B}$, this last condition holds if
> > > > > \begin{equation}
> > > > > T \geq  1 + \frac{1}{BA^2-1}.
> > > > > \end{equation}
> > > > > Again using our assumption $BA^2 \geq 2$ we have that $\frac{1}{BA^2-1} \leq 1$ thus it is sufficient to impose that $T \geq 2.$
> > > > > Substituting $\gamma = \frac{A}{\sqrt{T}}$ and the relaxation $\frac{1}{1-\theta^T} \leq 2$ into the original bound gives the claimed result.
> > > > >
> > > > > End of Proof.

---

> > > > > > ### Comment · Reviewer_gXey · 2023-08-16
> > > > > >
> > > > > > The result looks good to me. I'll increase the score then.

---

### Official Review · Reviewer_biC8 · 2023-07-08

**Soundness:** 3 good
**Presentation:** 3 good
**Contribution:** 2 fair
**Rating:** 7
**Confidence:** 2

**Summary:**

The paper proves 1/sqrt(T) respectively 1/T convergence rates for black-box variational inference methods, when implemented with a proximal stochastic gradient method.  Such rates were not available in the literature until now, due to difficulties in bounding the gradient noise. The main contribution of the paper is to prove new bounds on the gradient noise and incorporate them into the proof techniques for proximal SGD.

**Strengths:**

The paper is well-written and clearly states its contributions.  While the topic seems rather narrow, the technical part of the paper appears to be a novel and non-trivial result.

**Weaknesses:**

1) The paper solves a quite "narrow" problem, that is fixing some gaps in existing convergence proofs for variational inference. It is unclear whether this work will be of interest to the wider ICML community, as it is a niche topic, and the theory does not seem to directly suggest any practical improvements or changes.

2) BBVI is sometimes less preferable in practice than natural-gradient algorithms. These methods use the KL / Fisher-geometry (https://arxiv.org/abs/2107.04562). This line of works could be mentioned in the introduction.

3) There has been some recent works
https://arxiv.org/abs/2205.15902
https://proceedings.mlr.press/v202/diao23a.html
which prove convergence for BBVI-like algorithms by interpreting them as Wasserstein gradient flow. Perhaps these works could be mentioned.




**Questions:**

1) Which algorithm would be preferable in practice, prox-SGD or proj-SGD? From the theory side, it seems they have similar convergence rates, but maybe there are other things to take into consideration?


**Limitations:**

All limitations are addressed.

---

> ### Author Rebuttal · Authors · 2023-08-08
>
> Thank you for your review.
>
> ### Natural gradient algorithms
>
> We agree this should be discussed. We will also mention that extending theory to address these algorithms remains an open problem.
>
> ### Related work by Diao et al.
>
> We agree we should mention these recent works. (We note that they make use of Hessians which can be computationally challenging and is distinct from most practical BBVI algorithms.)
>
> ### Prefer prox-SGD or proj-SGD
>
> The reviewer is correct when saying that from a theory side, prox-SGD and proj-SGD have similar convergence rates with respect to T. If we want to compare them further, we need to look closely at the multiplicative constants in the rates, which depend essentially on the constants (a,b) in the quadratic bound. Since prox-SGD needs the “energy” estimator and proj-SGD needs the “entropy” estimator, the question could be shifted to: which estimator has the best constants (for instance the lowest variance). From that perspective, the two estimators we present have roughly the same constants.
>
> Another consideration in practice is the complexity of the projection vs proximal steps. In many cases, these will both be very cheap but in certain high-dimensional settings, the complexity of the proximal step could be higher (Θ(d^3) vs Θ(d^2), see lines 273-281).
>
> This being said, the story could be completely different when considering *other* estimators. Intuitively, we could imagine that an estimator will be able to have lower variance if it is a stochastic estimator of the full objective function instead of just the energy function. That scenario would give an advantage to the proj-SGD algorithm. We have just now been able to prove exactly this for the “Sticking-The-Landing (STL)” estimator. We showed that it is quadratically bounded, and further that its constant “b” is directly related to how close to a Gaussian the target is. This formalizes the idea that the easier the problem is (here approximating an almost Gaussian with a Gaussian), the lower the variance of the estimator is. In the extreme case that the target is exactly a Gaussian, this variance “b” is exactly zero, meaning that the rates drastically improve from 1/T to exponential rates. In this setting, proj-SGD would be a better choice.
> We propose to add this new small result in the paper, and to develop the discussion on the cost at the end of Section 4. We will underline the fact that the variance of the chosen estimator has a strong impact on the complexity (a well-known fact in stochastic optimization) and that some estimators can have a significantly smaller variance (like STL for easy problems).

---

> > ### Comment · Reviewer_biC8 · 2023-08-17
> >
> > Thanks for the detailed answer!
> >
> > Regarding Diao et al. necessarily requiring Hessians, this is not true. An expectation over Hessians can always be rewritten as expectation over gradients (via reparametrization trick).

---

### Official Review · Reviewer_6P4Y · 2023-07-15

**Soundness:** 3 good
**Presentation:** 3 good
**Contribution:** 3 good
**Rating:** 6
**Confidence:** 4

**Summary:**

The paper addresses the lack of provable convergence guarantees for black-box variational inference (VI) and proposes convergence guarantees for two stochastic optimization algorithms applied to Gaussian variational families.

The authors identify challenges in analyzing VI as a standard stochastic optimization problem, including the non-smoothness of the objective function, lack of uniform smoothness, and lack of uniformly bounded noise for gradient estimators.

They provide theoretical results and noise bounds for two gradient estimators and propose proximal and projected stochastic gradient descent algorithms.

I did not find technical flaws.

**Strengths:**

1. The paper addresses an important problem by providing provable convergence guarantees for black-box VI, which is widely used but lacks theoretical guarantees.

2. The authors clearly articulate the challenges in analyzing VI as a standard stochastic optimization problem, such as the non-smoothness of the objective function and the lack of uniform smoothness and noise bounds. The paper provides rigorous theoretical results and noise bounds for gradient estimators used in black-box VI.

3. The paper is well written. It provides a comprehensive and coherent review of the literature and clearly presents the technical contributions and proof ideas. I'm not a theory guy but I enjoy reading the paper.

**Weaknesses:**

1. I wonder if is it possible to perform some simple (even toy) examples to empirically check the results and illustrate the theory in a better way.

2. The dense Gaussian family is still limited (at least in my opinion). It is better to discuss whether the techniques developed in the paper can be applied to more general cases (e.g., black-box variational inference with a neural network model) or at least discuss the challenges.

**Questions:**

See weakness.

**Limitations:**

None.

---

> ### Author Rebuttal · Authors · 2023-08-08
>
> Thank you for your review. We will respond to two points.
>
> ### Examples
>
> We ask for some consideration for the constraints imposed by a 9-page limit. Given that this is a theoretical paper whose goal is to provide guarantees for algorithms already commonly used in practice, we prefer to focus entirely on theory.
>
> ### Dense Gaussians
>
> We agree this is an important issue that should be discussed more in the paper. We emphasize that our new optimization results in section 3 apply more broadly than just Gaussian VI (and indeed apply more broadly than VI)—they apply to any composite objective consisting of a smooth part and a non-smooth part where (1) Either (1a) The non-smooth part is proximalable or (1b) can be projected onto a constraint set where it is smooth and (2) the gradient noise obeys a quadratic bound. It is conceivable that such guarantees could be shown for more general variational families, and if so our convergence results would provide “plug in” optimization guarantees. We suspect it might be fairly easy to establish such guarantees for, e.g., elliptical distributions or location-scale families, using similar proof strategies as for Gaussians. But doing it for much broader classes would definitely be a challenge—the properties (1) and (2) we use above were developed over several years in a sequence of papers. We propose to add a short discussion of this topic to section 5 of the paper.

---

> > ### Comment · Reviewer_6P4Y · 2023-08-14
> >
> > Thanks for your reply. I'm happy for the clarifications regarding how the theory applies to more general cases. I will keep my score but will increase my confidence.

---

### Official Review · Reviewer_TUBr · 2023-07-27

**Soundness:** 3 good
**Presentation:** 3 good
**Contribution:** 3 good
**Rating:** 5
**Confidence:** 3

**Summary:**

This paper analyzed the convergence of Black-box VI, which has been widely used in variational inference in recent years. Under the assumption that the target joint distribution is convex or strongly convex and the variational posterior distribution is Gaussian, the convergence of the variational parameter obtained by Black-box VI to the ELBO optimal solution is first presented when using Prox-GD and Projected Gradient Descent. An important aspect of the analysis of Black-box VI is that the gradient is stochastic.
To deal with this, the authors introduced a class of quadratically bounded estimators as estimators of the gradient and combined them appropriately with the existing Prox-GD analysis.



**Strengths:**

Although the proposed analysis makes a number of assumptions, the results are significant in that they include an important Bayesian model that is used in practice.

For the first time, an analysis of the convergence of Black-box VI is presented, albeit with a rather limited problem, such as the assumption of variational posterior distribution as Gaussian distribution.

**Weaknesses:**

It was very difficult to understand how novel and important the proposed analysis is from a single reading. In existing studies, each component of the presented analysis is already well known, but I am not sure how important and novel the presented  VI analysis is as the combination of those well-known results.

By assumption, since the objective function is convex (or strongly convex), the convergence using Prox SGD or other methods seems apparent, the difficulty of the problem stems from the noisy version of the gradient, as discussed in  Sec 2.3. On the other hand, as there have been many studies discussing noisy versions of prox GD, such as Stochastic Proximal Gradient Descent[1], so I am not sure how novel this analysis is.

In the proofs of Theorem 6 and 7, I understood that the novelty is mainly the technique to treat the gradient estimator that satisfies the quadratic bounded property shown in Definition 4. However, I still did not understand how important it is compared to existing studies such as the Stochastic Proximal Gradient Descent study.

[1]Convergence of Stochastic Proximal Gradient Algorithm, L. Rosasco et al.



**Questions:**

I would like to know more about the novelty of this study in the context of Stochastic Proximal Gradient Descent in terms of technical aspects and assumptions of the proof.

**Limitations:**

Yes, the limitations of the statements are clearly explained.

---

> ### Author Rebuttal · Authors · 2023-08-08
>
> Thank you very much for your review.
>
> ### Novelty of the analysis
>
> Given the current state of knowledge, the convergence of Prox/Proj-SGD is not apparent, because the convexity of the objective function alone is not enough. A second essential hypothesis must be satisfied by the estimator of the gradient: typically, results require a noise bound that depends on the suboptimality gap, gradient norms and constants. This is for instance the case in the paper [1] suggested by the reviewer, and we also refer to [KR] for a good discussion on noise bounds that result from standard assumptions. Those standard noise bounds can be verified under the common assumption that the stochastic functions are (uniformly) smooth, or Lipschitz. But our problem is that our gradient estimators do not satisfy such a typical noise bound, but instead the quadratic error bound presented in the paper. There is indeed a significant literature on the analysis of Prox/Proj-SGD, but we carried out a thorough bibliographic review and consulted with experts to find that no existing analysis could accommodate such quadratic noise bound while at the same time handling a composite non-smooth objective. The goal of Section 2.3 in our paper is to explain to the reader exactly what in our paper is novel. We will try to explain this more clearly in the revision.
>
> [KR] Better Theory for SGD in the Nonconvex World, Ahmed Khaled & Peter Richtarik, TMLR 2022
>
> ### Is the main novelty in treatment of the gradient estimators?
>
> Broadly speaking, yes. Our proof technique borrows several ingredients from previous analyses of SGD, and the way we handle convexity and smoothness is rather classical. The main difficulty we faced is how to handle the quadratic noise bound together with composite non-smooth objective.. For this we needed a relatively new weighted telescoping technique, the application of which involves substantial technical difficulties (see proofs on pages 21-30 of appendix).

---

> > ### Comment · Reviewer_TUBr · 2023-08-18
> >
> > Thank you for the detailed clarification. My concerns are now solved. I will increase the contribution score.

---

### Decision · Program_Chairs · 2023-09-21

**Decision:**

Accept (poster)

**Comment:**

This paper proved $1/\sqrt{T}$ and $1/T$ convergence rate for black-box VI with convex and strongly convex model log likelihood, respectively, under Gaussian variational distribution assumption and some Lipschitz conditions for the model. The main techniques are to provide control for the noise in path-based BBVI gradient estimators as well as to use theoretical results in stochastic proximal gradient descent in some smart way.

Reviewers commend that the paper, although with assumptions that narrow the scope, makes a significant progress towards addressing the important theoretical challenge of convergence guarantees for BBVI. They provided comments regarding discussions with existing theoretical results analysing Gaussian VI as well as another very recent work [Xu & Campbell, 2022] related to BBVI theoretical studies. One reviewer also pointed out a bug in one of the lemma, which is then fixed by the authors without impacting on the main theoretical conclusions.

One reviewer mentioned potentially adding experiments to verify the theory. I personally think that experiments are nice to have, but given the significant contribution on the theory front, having no experiment shouldn't be a reason for rejection in this case.

I suggest the authors to include discussions regarding other theoretical papers for Gaussian VI and [Xu & Campbell 2022] in revision.